# A comprehensive survey of *C. elegans* argonaute proteins reveals organism-wide gene regulatory networks and functions

Uri Seroussi, Andrew Lugowski, Lina Wadi, Robert X Lao, Alexandra R Willis, Winnie Zhao, Adam E Sundby, Amanda G Charlesworth, Aaron W Reinke, Julie M Claycomb*

Department of Molecular Genetics, University of Toronto, Toronto, Canada

**Abstract** Argonaute (AGO) proteins associate with small RNAs to direct their effector function on complementary transcripts. The nematode *Caenorhabditis elegans* contains an expanded family of 19 functional AGO proteins, many of which have not been fully characterized. In this work, we systematically analyzed every *C. elegans* AGO using CRISPR-Cas9 genome editing to introduce GFP::3xFLAG tags. We have characterized the expression patterns of each AGO throughout development, identified small RNA binding complements, and determined the effects of *ago* loss on small RNA populations and developmental phenotypes. Our analysis indicates stratification of subsets of AGOs into distinct regulatory modules, and integration of our data led us to uncover novel stress-induced fertility and pathogen response phenotypes due to *ago* loss.

## Editor's evaluation

This impressive study presents the most comprehensive analysis of the Argonautes, their small RNA partners, their targets, and their biological functions in any species to date. The work provides new insights into Argonaute-based pathways, includes extensive validation of existing models, and describes overall a treasure-trove of reagents and datasets for future exploration of the vast Argonaute world in *C. elegans*.

*For correspondence: julie.claycomb@utoronto.ca

## Introduction

Small RNA-mediated gene regulatory pathways (collectively referred to as RNA interference [RNAi]) have been identified in organisms from all domains of life (*Swarts et al., 2014*). These pathways utilize an array of molecular mechanisms in the epigenetic modulation of gene expression and exert their influence on nearly every step in the lifecycle of a transcript, from transcription to translation (*Meister, 2013*; *Wu et al., 2020*). At the cellular level, small RNA (sRNA) pathways are key contributors to regulating genome and transcriptome homeostasis, both under normal conditions and stress responses. At the organismal level, sRNA pathways are key regulators of gene expression programs that direct development and differentiation, and mis-regulation of sRNA pathways or components can lead to conditions such as cancer and infertility (*Wu et al., 2020*).

The central effectors of sRNA pathways are the highly conserved Argonaute (AGO) family of proteins. AGOs are the core components of ribonucleoprotein complexes called RISCs (RNA induced silencing complexes) and are guided in a sequence-specific manner by sRNAs (18–30 nucleotides long) to complementary target transcripts (*Dueck and Meister, 2014*). AGOs have a bilobed structure

consisting of four major domains: PAZ, MID, PIWI, and a low-complexity N-terminal domain. The PAZ and MID domains possess pockets to coordinate 3′ and 5′ end sRNA binding, respectively (*Sheu-Gruttadauria and MacRae, 2017*). The PIWI domain resembles RNaseH and has the capacity to direct endonucleolytic cleavage of the target RNA if the active site harbors a tetrad of catalytic amino acids (DEDD/H, *Nakanishi et al., 2012*). Relatively few AGOs possess this catalytic tetrad, and many AGOs recruit additional proteins to elicit other gene regulatory outcomes, such as mRNA de-capping, de-adenylation, or chromatin modulation.

In *Caenorhabditis elegans*, at least four types of endogenous sRNAs—miRNAs, piRNAs, 22G-RNAs, and 26G-RNAs—and as many as 27 AGO-like genes have the potential to contribute to complex networks of gene regulation in different tissues throughout development. miRNAs and piRNAs are genomically encoded and transcribed by RNA polymerase II, while the 22G-RNAs and 26G-RNAs are generated by the activity of different RNA-dependent RNA polymerases (RdRPs). miRNAs are known to associate and function with the conserved AGOs ALG-1, ALG-2, and ALG-5 (*Hutvagner et al., 2004*; *Brown et al., 2017*; *Corrêa et al., 2010*; *Vasquez-Rifo et al., 2013*). The Piwi-interacting RNAs (piRNAs, also called 21U-RNAs in *C. elegans*) bind to the PIWI AGO PRG-1 and are thought to maintain germline genome integrity by silencing foreign or deleterious nucleic acids such as transgenes (*Lee et al., 2012*; *Shirayama et al., 2012*). Although piRNA pathways in other animals play a more prominent role in regulating transposable elements than in *C. elegans*, the functions of the piRNA pathway are broadly and consistently required in animal germlines to ensure fertility (*Ozata et al., 2019*).

Two additional types of endogenous sRNAs present in *C. elegans* are the 26G-RNAs and 22G-RNAs (named for their predominant length and 5′ nucleotide). Because these sRNAs are generated by RdRPs, they are thought to exploit perfect complementarity to their targets. The 26G-RNAs are synthesized by the RdRP RRF-3, which generates dsRNA that is processed into 26G-RNAs by the endonuclease DICER and the phosphatase PIR-1 (*Chaves et al., 2021*). 26G-RNAs are classified into two groups: those of spermatogenic origin (class I, associated with ALG-3 and ALG-4, *Han et al., 2009*; *Conine et al., 2010*) and those of oogenic and embryonic origin (class II, associated with ERGO-1, *Vasale et al., 2010*; *Han et al., 2009*).

The 22G-RNAs are generated by the RdRPs RRF-1 and EGO-1, independent of DICER. Currently, 22G-RNAs are divided into two main groups, those that are bound by CSR-1 and target germline expressed protein-coding transcripts to protect them from silencing, along with fine-tuning gene expression in the embryo (*Claycomb et al., 2009*; *Cecere et al., 2014*; *Seth et al., 2013*; *Wedeles et al., 2013*; *Singh et al., 2021*; *Gerson-Gurwitz et al., 2016*; *Nguyen and Phillips, 2021*; *Charlesworth et al., 2021*), versus those that are bound by other WAGO class AGOs (such as WAGO-1 and HRDE-1) that silence protein-coding genes, pseudogenes, transposable elements, and cryptic loci (*Gu et al., 2009*). 22G-RNAs are generally thought to act as secondary, amplified sRNAs that are synthesized after a transcript is targeted by a primary sRNA/AGO complex, with the main exception being the majority of CSR-1 associated 22G-RNAs. Primary sRNAs take several forms: piRNAs (PRG-1), 26G-RNAs (ALG-3, ALG-4, and ERGO-1), and exogenous-siRNAs produced by DICER during exogenous RNAi (exoRNAi) (RDE-1).

*C. elegans* AGOs have generally been studied on a case-by-case basis, with *agos* being uncovered via genetic screens, or selected for study based on phenotype (e.g., *Tabara et al., 1999*). Such approaches are limited because not all phenotypes to which *agos* may contribute have been tested, and redundancy among the *agos* could confound their recovery in genetic screens. To date, only one study has taken a systematic approach to understanding AGO functional relationships, examining the requirement for each *ago* in exogenous RNAi (*Yigit et al., 2006*). A lack of antibodies against individual AGOs and difficulties with transgenic approaches have also hampered the development of a cohesive set of reagents to study AGO function. Indeed, several *C. elegans* AGOs have yet to be studied, and others remain only partially characterized.

In this study, we have undertaken a systematic analysis of the *C. elegans* AGOs. We employed CRISPR-Cas9 genome editing to introduce GFP::3xFLAG epitope tags in the endogenous loci of each *ago* using these strains to examine spatiotemporal expression profiles throughout development, combined with sequencing sRNAs from AGO complexes and *ago* mutants to define the core of *C. elegans* sRNA pathways. We systematically assessed fertility of *ago* mutants and employed phenotypic assays directed by our expression and sRNA sequencing data, enabling us to uncover new roles

for specific AGOs in maintaining germline integrity and in regulating immune responses to bacterial and viral pathogens. Collectively, our findings provide a foundation for understanding the full scope of sRNA pathway activity in *C. elegans*. With these AGO tools and knowledge of sRNA binding partners and targets, our findings provide a deeper understanding of sRNA functions throughout development and under varied environmental conditions in *C. elegans*.

## Results

### Systematic analysis of *C. elegans* Argonautes

Previous studies identified 27 *ago* genes in *C. elegans* (*Yigit et al., 2006*); however, some have been reclassified as pseudogenes (e.g., *prg-2*). To define an updated set of *ago* genes to study, we searched the genome (WormBase version WS262) for genes that contain PAZ and PIWI domains, have a predicted protein size of ~100 kDa, and bear homology to known AGOs. Twenty-one genes met these criteria, and we ultimately characterized 19 of these AGOs (*Supplementary file 1*, *Figure 1A and B*). Construction of a phylogenetic tree for these 19 AGOs in relation to *Arabidopsis thaliana* AGO1 and *Drosophila melanogaster* PIWI places seven AGOs in the AGO clade: ALG-1, ALG-2, ALG-3, ALG-4, ALG-5, ERGO-1, and RDE-1; a single AGO in the PIWI clade: PRG-1; and 13 AGOs in the WAGO clade: CSR-1, C04F12.1 (renamed VSRA-1 for Versatile Small RNAs Argonaute-1, see below), WAGO-1, PPW-2/WAGO-3, WAGO-4, SAGO-2/WAGO-6, PPW-1/WAGO-7, SAGO-1/WAGO-8, HRDE-1/WAGO-9, WAGO-10, and NRDE-3/WAGO-12 (*Figure 1A*). It is important to note that CSR-1 exists as two isoforms, a long isoform (CSR-1a) and a short isoform (CSR-1b), that differ by 163 amino acids at their N terminus (*Nguyen and Phillips, 2021*; *Charlesworth et al., 2021*). Here, we generally use 'CSR-1' to refer to both isoforms of the protein and designate specific isoforms in the text and figures as relevant.

We used CRISPR-Cas9 genome editing to introduce a GFP::3xFLAG tag into the N-terminus or within the first exon of the endogenous gene loci of all 21 *agos* (*Dickinson et al., 2015*; *Figure 1—figure supplement 1A*). We detected GFP expression for both the transcriptional reporter and GFP::3xFLAG::AGOs for 19 AGOs by confocal microscopy. (Note that both isoforms of CSR-1 were tagged in the strain used here, JMC101.) We verified that the GFP::3xFLAG::AGOs were full-length proteins by Western blot analysis (*Figure 1—figure supplement 1B*). We were unable to detect WAGO-5 and WAGO-11 fusion protein expression by microscopy or Western blot analysis under common laboratory conditions (*Figure 1—figure supplement 2A*). RNA-tiling-array data (*Celniker et al., 2009*) showed that both *wagos* are expressed at low levels with poor sequencing coverage (*Figure 1—figure supplement 2B*), and RT-PCR only detected low amounts of *wago-11* mRNA (*Figure 1—figure supplement 2C*). *wago-5* is targeted by WAGO-1-associated 22G-RNAs, suggesting it is silenced by the WAGO pathway (*Figure 1—figure supplement 2D*). During this project, the designation of *wago-11* was changed to 'pseudogene' in WormBase (version WS275). Our data support that these *wagos* are pseudogenes; therefore we excluded these WAGOs from further analysis.

We tested the function of the tagged AGOs by assessing phenotypes associated with *ago* loss of function (*Yigit et al., 2006*; *Batista et al., 2008*; *Guang et al., 2008*; *Han et al., 2009*; *Buckley et al., 2012*; *Brown et al., 2017*; *Xu et al., 2018*; *Figure 1C–H*, Figure 6D, and E, Figure 7D, and E). While previous studies have defined loss-of-function defects for several *agos*, some had no known loss-of-function defects (*Supplementary file 1*). Of all the AGOs tested, the GFP::3xFLAG tagged AGOs WAGO-1 and NRDE-3 did not behave as wild-type in phenotypic assays (*Figure 1C and H*). Therefore, we tagged WAGO-1 and NRDE-3 with only 3xFLAG (*Figure 1—figure supplement 2E*) for the purpose of small RNA cloning and used the GFP::3xFLAG tagged strain for analysis of expression patterns. Both 3xFLAG tagged AGOs were functional in phenotypic assays (*Figure 1C and H*). We note that although PPW-2 (also known as WAGO-3) and WAGO-1 were recently shown to be N-terminally processed by the protease DPF-1, both GFP::3xFLAG::PPW-2 and 3xFLAG::WAGO-1 strains behaved as wild-type, consistent with previous reports (*Gudipati et al., 2021*; *Schreier et al., 2022*). The tagged C04F12.1/VSRA-1 and WAGO-10 were not tested for a specific phenotype as there are no known phenotypes for *C04F12.1/vsra-1* and *wago-10* mutants. As we observed that the GFP tag may interfere with function in some instances (e.g., WAGO-1 and NRDE-3), we tagged C04F12.1/VSRA-1 and WAGO-10 with 3xFLAG at the same position as the GFP::3xFLAG tags for the purpose of sRNA cloning out of an abundance of caution (*Figure 1—figure supplement 2E*).

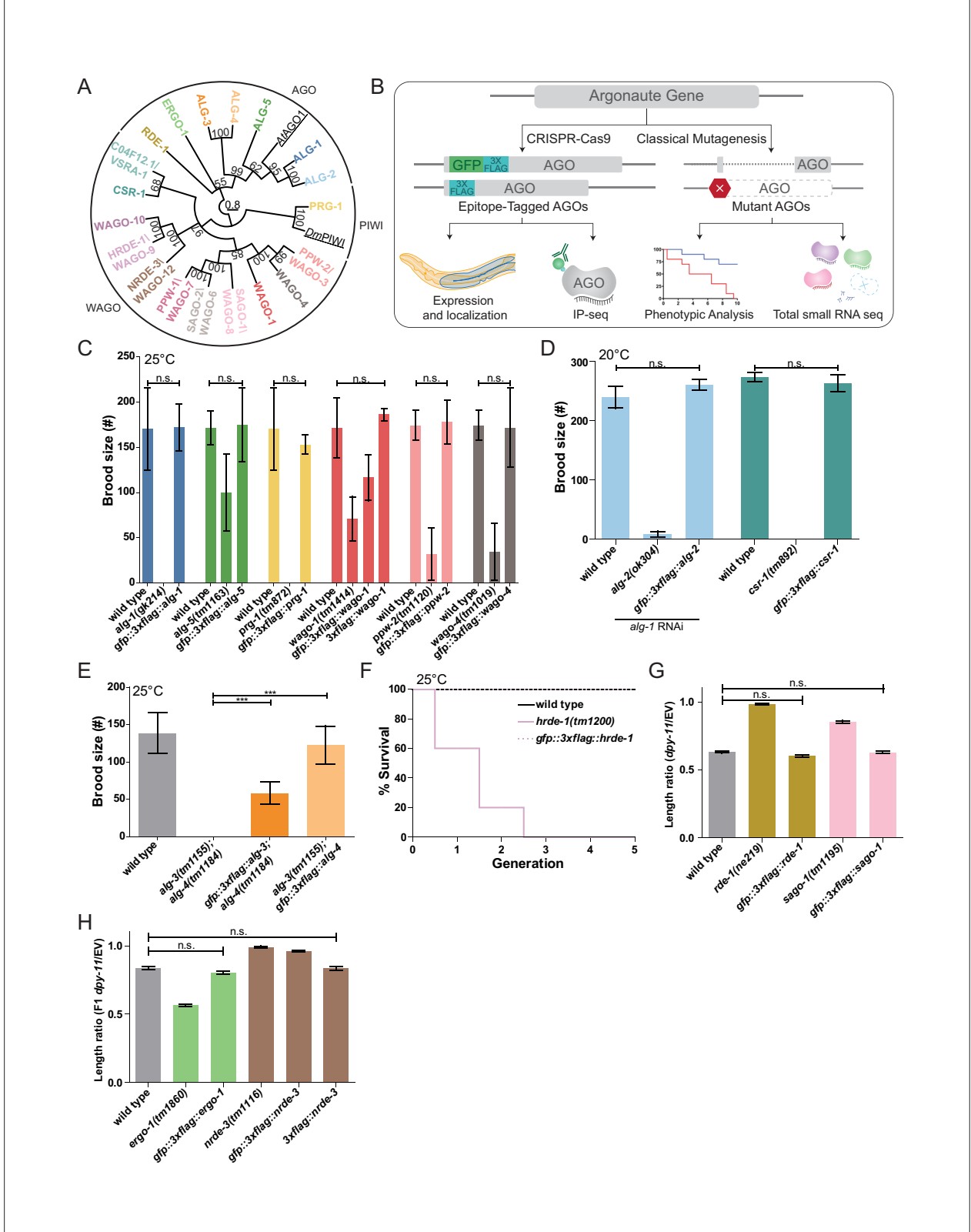

**Figure 1.** Functional validations of GFP::3xFLAG and 3xFLAG tagged Argonautes. (**A**) Maximum likelihood evolutionary tree of *A. thaliana* AGO1 (AtAGO1), *D. melanogaster* PIWI (DmPIWI), and *C. elegans* Argonautes. (**B**) Workflow for characterizing *C. elegans* Argonautes. (**C**) Functional validations of tagged ALG-1, ALG-5, PRG-1, WAGO-1, PPW-2, and WAGO-4 strains. Brood size was determined at 25°C for each indicated genotype. N ≥ 5 worms per condition. (**D**) Functional validation of tagged ALG-2 and CSR-1 strains. Brood size was determined at 20°C for each indicated genotype.

*Figure 1 continued on next page*

*Figure 1 continued*

For the ALG-2 tag validations, the brood size was determined when worms were fed dsRNA of *alg-1*. N = 5 worms per condition. (**E**) Functional validations of tagged ALG-3 and ALG-4 strains. Brood size was determined at 25°C for each indicated genotype. N = 10 worms per condition. (**F**) Functional validation of the tagged HRDE-1 strain via a Mortal germline assay at 25°C. N = 5 worms per condition. (**G**) Functional validations of tagged RDE-1 and SAGO-1 strains. Worms were fed bacteria expressing dsRNA of *dpy-11* or an empty vector (EV) RNAi control. The length ratio of *dpy-11* dsRNA fed P0 worms compared to the average length on EV was determined. N = 30 worms per condition. (**H**) Functional validations of tagged ERGO-1 and NRDE-3 strains. Worms were fed bacteria expressing dsRNA of *dpy-11* or an EV control. The length ratio of the F1s of the *dpy-11* dsRNA fed worms compared to the average length on EV was determined. N = 30 worms per condition. (**C–H**) *p-value<0.05, **p-value<0.01, ***p-value, n.s. = not significant. One-way ANOVA with Tukey's post hoc multiple comparison test. All error bars represent standard deviation.

The online version of this article includes the following source data and figure supplement(s) for figure 1:

**Figure supplement 1.** Epitope tag locations and Western blot validations.

**Figure supplement 1—source data 1.** This file contains original western blots of AGO IPs used in creating *Figure 1—figure supplement 1*.

**Figure supplement 2.** Additional analyses of tagged Argonautes.

**Figure supplement 2—source data 1.** This file contains original western blots of AGO IPs used in creating *Figure 1—figure supplement 2*.

## AGO sequencing reveals sRNA association

The sRNAs associated with each AGO provide important insight into the transcripts the AGOs may regulate. To identify the sRNAs that interact with each AGO, we performed immunoprecipitation (IP) followed by high-throughput sequencing of sRNAs for each of the tagged AGOs in duplicate. In parallel, we sequenced total sRNAs from the same lysates as the IPs ('Input' samples). We conducted IPs on worm populations at the L4 to young adult transition (58 hr post L1 synchronization), because all but a few AGOs are expressed at this stage. The only exceptions were ALG-3, ALG-4, and WAGO-10, which are only expressed during spermatogenesis (*Charlesworth et al., 2021*). Therefore, we conducted ALG-3, ALG-4, and WAGO-10 IPs during the mid L4 stage (48 hr post L1 synchronization), during which spermatogenesis occurs. For consistency, we treated all libraries with 5′ polyphosphatase to enable detection of 5′ tri-phosphorylated small RNA species (22G-RNAs), along with 5′ mono-phosphorylated species (miRNAs, piRNAs, 26G-RNAs) (*Supplementary file 2*).

We assessed the length and 5′ nucleotide distribution of sRNAs associated with each AGO and mapped this total set of reads to genomic features (*Figure 2A*). We also determined how many miRNAs, piRNAs, and other genomic features (protein-coding genes, transposable elements, pseudogenes, and long intergenic noncoding RNAs, or lincRNAs that are targeted by antisense endogenous siRNAs including, but not limited to, 22G-RNAs and 26G-RNAs) were *enriched* over twofold in both IP replicates relative to the input samples (*Supplementary file 3*). For this enrichment analysis, we did not place any constraints on sRNA length or 5′ nucleotide and considered all genome mapping antisense reads. We defined 22G-RNA reads as 20-24 nt with no 5′ nucleotide bias and 26G-RNAs as 25-27 nt with no 5′ nucleotide bias. These stringent criteria led to the assignment of a high-confidence set of AGO-enriched sRNAs. We refer to transcripts for which antisense siRNAs are enriched over twofold as the 'targets' of AGO/sRNA complexes.

A subset of the AGO clade primarily associated with miRNAs (*Corrêa et al., 2010*; *Vasquez-Rifo et al., 2013*; *Brown et al., 2017*). For ALG-1, miRNAs comprised ~98% of all associated reads (47 miRNAs enriched); ALG-2, ~98% (81 enriched), ALG-5, ~68% (37 enriched); and RDE-1 (103 enriched). RDE-1 is involved in exoRNAi (*Tabara et al., 1999*); however, we observed that it also associates with 22G-RNAs targeting protein-coding genes (~21%, 536 enriched genes). We expect that these 22G-RNAs possess a 5′ mono-phosphate, given that RDE-1 was previously shown to associate with multiple types of sRNAs that are likely DICER products (*Corrêa et al., 2010*). We also detected abundant miRNAs associated with the rest of the AGO clade AGOs ALG-3, ALG-4, and ERGO-1, with ~55% (26 enriched), ~40% (2 enriched), and ~26% (33 enriched) of reads corresponding to miRNAs, respectively. These three AGOs were previously described as genetically required for 26G-RNA accumulation, and ERGO-1 was shown to physically associate with 26G-RNAs (*Conine et al., 2010*; *Vasale et al., 2010*). Indeed, ALG-3, ALG-4, and ERGO-1 all associate with 26G-RNAs with ~17, 32, and 15% of reads corresponding to 26G-RNAs respectively (*Figure 2A and B*). The endo-siRNAs in ALG-3 and ALG-4 IPs are primarily antisense to protein-coding genes (~22%, 2561 enriched and ~39%, 2848 enriched, respectively) and pseudogenes (~1%, 66 enriched and 1.2%, 97

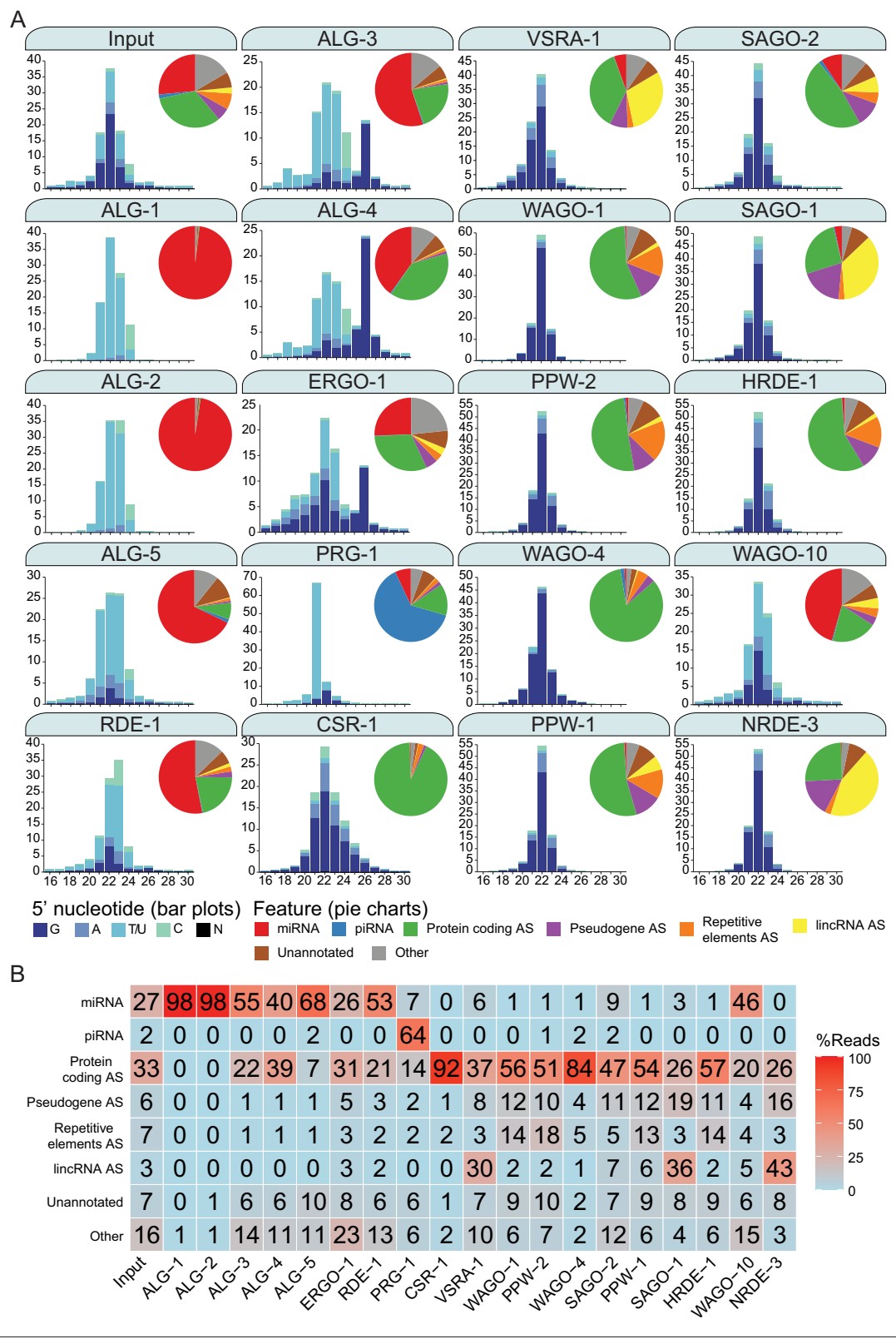

**Figure 2.** Argonautes associate with different types of sRNAs and target different categories of genetic features. (**A**) 5' nucleotide and length of sRNAs present in each Argonaute IP shown in bar graph form. The pie charts depict which type of genetic element (biotype) the sRNAs correspond to, as listed. AS = antisense, S = sense. The 'Other' category encompasses: miRNA AS, piRNA AS, protein-coding S, pseudogene S, repetitive elements S, lincRNA S, rRNA S/AS, snoRNA S/AS, snRNA S/AS, tRNA S/AS, ncRNA S/AS, and antisense lncRNAs (ancRNAs/*anr* loci) (***Nam and Bartel, 2012***).

*Figure 2 continued on next page*

Figure 2 continued

The average of two biological replicates is shown. The GFP::3xFLAG tagged Argonautes were used for IPs except for C04F12.1/VSRA-1, WAGO-10, and NRDE-3, where a 3xFLAG tag was used. All IPs were performed on Young Adult samples except for ALG-3, ALG-4, and WAGO-10, which were conducted on L4 staged animals. The CSR-1 strain tags both isoforms. (**B**) A table summarizing the percentage of reads in each set of AGO IPs corresponding to genetic element types in (**A**).

enriched, respectively), while ERGO-1 targets primarily protein-coding genes (~31%, 411 enriched), pseudogenes (~5%, 39 enriched), and lincRNAs (~3%, 30 enriched) (*Figure 2A and B*).

PRG-1, the only PIWI homolog in *C. elegans*, PRG-1, is known to associate with, and be required for piRNA stability (*Wang and Reinke, 2008*; *Batista et al., 2008*; *Das et al., 2008*). It is the major piRNA-associated AGO, and ~64% of the reads in PRG-1 IPs correspond to piRNAs (5932 enriched) (*Figure 2A and B*). We also detected 22G-RNAs (~21%) that primarily targeted protein-coding genes enriched in PRG-1 complexes (150 gene targets enriched).

Previous studies predicted that all WAGOs would associate with 22G-RNAs (*Guang et al., 2008*; *Gu et al., 2009*; *Buckley et al., 2012*; *Xu et al., 2018*), and we observed that 22G-RNAs are the most abundant class of small RNAs associated with the WAGOs, many of which are antisense to protein-coding genes, ranging from ~26% of reads in SAGO-1 IPs to ~93% of reads in CSR-1 IPs. Groups of WAGOs associate more prominently with sRNAs that target specific genomic features including pseudogenes, lincRNAs, and repetitive and transposable elements. Pseudogenes are primarily targeted by HRDE-1 (~19% of reads in the IP are antisense to these elements), NRDE-3 (~17%), WAGO-1 (~12%), PPW-1 (~12%), SAGO-2 (~11%), SAGO-1 (~11%), and PPW-2 (~10%) (*Figure 2A and B*). lincRNAs are primarily targeted by NRDE-3 (~43% of reads in the IP are antisense to these elements), SAGO-1 (~35%), C04F12.1/VSRA-1 (~30%), SAGO-2 (~7%), and PPW-1 (~6%) (*Figure 2A and B*). sRNAs antisense to repetitive and transposable elements are most abundant in PPW-2 (~19% of reads in the IP are antisense to these elements), WAGO-1 (~18%), HRDE-1 (~15%), and PPW-1 (~14%) complexes (*Figure 2A and B*).

## miRNAs

### miRNAs associate with ALG-1, ALG-2, ALG-5, and RDE-1

A total of 437 miRNAs are annotated by mirBase 22.1 (encompassing 253 families with individual 5p or 3p strands). We detected reads for 402 miRNAs across our AGO Input and IP samples. Of these, 190 were found to be enriched over twofold in the AGO IPs. We observed miRNAs in association with the known miRNA binding AGOs, ALG-1, ALG-2, ALG-5, and RDE-1, and in association with the 26G-RNA binding AGOs ALG-3, ALG-4, and ERGO-1 (*Figure 3A*). Of the miRNAs enriched in the AGO IPs, some were enriched in association with only one AGO while others were enriched in multiple AGOs (*Figure 3A*). For example, 103 miRNAs were enriched in RDE-1 complexes, with 55 being exclusive to RDE-1. RDE-1 was the only AGO where the majority (63/103) of enriched miRNAs were not conserved with those of the related nematode, *Caenorhabditis briggsae*, suggesting that newly evolved miRNAs may be routed initially into the RDE-1 pathway before subsequently integrating into the ALG-1/2/5 pathway (*Figure 3L*).

Adjusting the number and position of mismatches in the precursor miRNA (pre-miRNA) duplex of a transgenic *let-7* miRNA has been shown to shift the balance between ALG-1 and RDE-1 loading of the transgenic *let-7*. These experiments demonstrated that ALG-1 preferentially associates with miRNAs from mismatched precursors while RDE-1 prefers perfectly matching precursors (*Steiner et al., 2007*). Therefore, we examined the number of mismatches in precursor miRNA duplexes that are enriched in ALG-1, ALG-2, ALG-5, and RDE-1 complexes (*Figure 3B*). miRNAs loaded into RDE-1 showed a lower average of mismatches in their precursors, and miRNAs derived from precursors with no mismatches were only bound by RDE-1, although some miRNAs derived from mismatched precursors were also loaded into RDE-1. These data suggest that endogenous miRNAs with higher complementarity in their precursor duplex are preferentially loaded into RDE-1.

Given that ALG-1 and ALG-2 are thought to be required for the stability of miRNAs (*Brown et al., 2017*), and sRNAs are generally unstable in the absence of their AGO binding partner, we asked whether these AGOs are required for the stability of their enriched miRNAs or for miRNAs in general, by sequencing sRNAs from *ago* mutants and wild-type worms. We found that the AGO-enriched miRNAs were substantially depleted in *alg-1*, *alg-2*, *alg-5*, and *rde-1* mutants (*Figure 3C*), and five

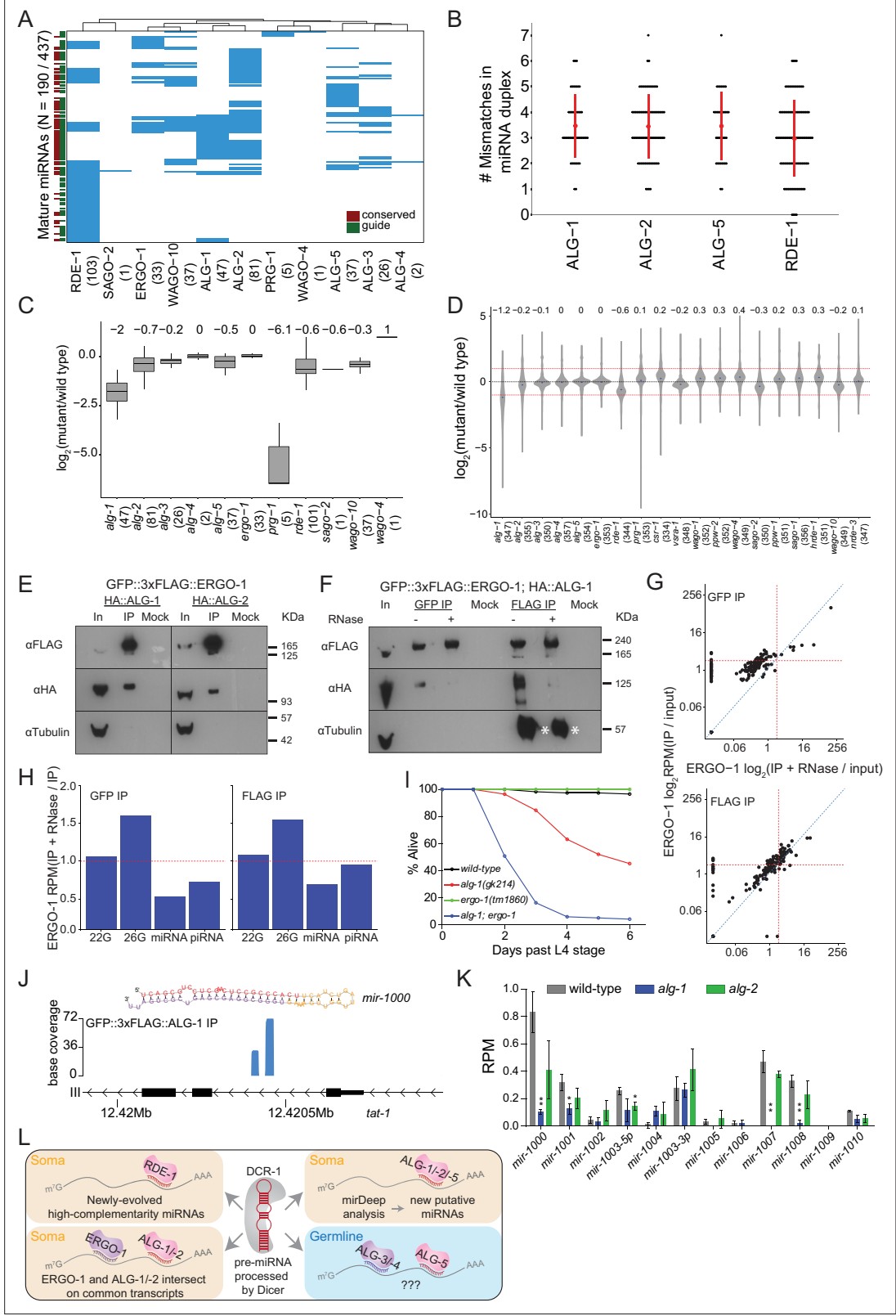

**Figure 3.** Analysis of miRNAs in AGO IPs and *ago* mutants reveals novel miRNAs. (**A**) Clustering diagram of miRNAs enriched in AGOs. Each blue line represents an individual mature miRNA sequence. miRNAs are categorized by conservation, with those conserved to *C. briggsae* designated in burgundy on the left, and by whether they are the canonical guide strand, as designated in green on the left. (**B**) The number of mismatches in the precursor miRNA sequences for which a mature miRNA was enriched in the indicated AGOs. Each black dot represents a single precursor miRNA

*Figure 3 continued on next page*

*Figure 3 continued*

duplex. The red dot and lines indicate the average and standard deviation. (**C**) Fold change of miRNAs enriched in AGO IPs in the corresponding *ago* mutant. (**D**) Fold change of all detected miRNAs in *ago* mutants compared to wild-type. (**E**) Western blots of co-IP experiments of ERGO-1 and ALG-1 and ALG-2. GFP::3xFLAG::ERGO-1 was crossed to HA::ALG-1 or HA::ALG-2 strains and IPed using anti-GFP antibodies. (**F**) As in (**E**) but GFP::3xFLAG::ERGO-1 IPs were conducted either with anti-GFP or anti-FLAG antibodies with or without RNase treatment. Asterisks indicate IgG. (**G**) Scatter plots showing enrichment of miRNAs in IP and IP +RNase-treated samples of GFP::3xFLAG::ERGO-1. The top graph shows the results of an anti-GFP IP and the bottom graph shows the results of an anti-FLAG IP (one replicate per condition). (**H**) Bar plots showing quantification of sRNA types in IP and IP +RNase-treated samples of GFP::3xFLAG::ERGO-1 (anti-GFP IP on the left; anti-FLAP IP on the right). (**I**) Survival of worms of the indicated genotype beyond the L4 stage (bottom). N ≧ 100. (**J**) An example of a novel miRNA within an intron of the gene *tat-1* as determined by mirDeep2 analysis of ALG-1 IPs. (**K**) Analysis of the levels of predicted novel miRNAs in wild-type, *alg-1* and *alg-2* mutants. Predicted novel miRNAs are provisionally named. Error bars represent standard deviation. (**L**) A summary of miRNA pathway observations.

The online version of this article includes the following source data for figure 3:

**Source data 1.** This file contains original western blots of AGO IPs used in creating **Figure 3**.

PRG-1-enriched 'miRNAs' were over 60 times lower in abundance in *prg-1* mutants (see below). Loss of ALG-1 had the most substantial effect on global miRNA levels, leading to a greater than twofold decrease (**Figure 3D**), indicating that *alg-1* is genetically required for the stability of most miRNAs and potentially explaining why *alg-1* mutants have more severe phenotypes than the other miRNA-associated *ago* mutants (**Bukhari et al., 2012**; **Brown et al., 2017**). Loss of RDE-1 also led to a substantial depletion of miRNAs overall, while loss of ALG-2 or ALG-5 did not result in major changes, which could reflect redundancy or differences in function for these AGOs. Loss of several WAGOs also led to a decrease in global miRNA levels, and we speculate that this is due to indirect effects on protein-coding gene regulation, rather than a direct influence on the miRNA pathway.

## miRNA and ERGO-1 26G-RNA pathways intersect

To understand the association of ALG-3, ALG-4, and ERGO-1 with both 26G-RNAs and miRNAs, we tested whether the GFP::3xFLAG tag interfered with proper sRNA loading, using existing ERGO-1 IP-sRNA sequencing data performed with an ERGO-1-specific antibody (**Vasale et al., 2010**). In this study, the authors detected a subset of ERGO-1-associated miRNAs, but dismissed this as a nonspecific interaction. We reanalyzed these data using our custom computational pipeline and identified 26 miRNAs that were enriched twoold over input and significantly overlapped with our IP data set, suggesting that miRNA association with ERGO-1 is not a result of the GFP::3xFLAG tag, but that miRNA enrichment may be a property of ERGO-1 IPs.

We examined our AGO expression and localization data, and observed that ERGO-1 expression closely overlaps with ALG-1 and ALG-2, indicating that these AGOs could physically interact in vivo (Figures 6A and F and 7A, **Figure 7—figure supplements 1–8**). To determine whether ERGO-1 physically interacts with one or more of the miRNA-binding AGOs, we crossed GFP::3xFLAG::ERGO-1 worms to HA::ALG-1 and HA::ALG-2 tagged strains (**Brown et al., 2017**) and performed co-IP experiments. We found that ERGO-1 physically interacted with both ALG-1 and ALG-2 (**Figure 3E**) in an RNA-dependent manner (**Figure 3F**). These data suggest that the ERGO-1/ALG-1 or -2 interactions are due to AGO associations on shared target transcripts. Further supporting this model, we sequenced sRNAs associated with ERGO-1 IPs after RNase treatment and observed that miRNAs were substantially reduced, while 26G-RNAs remained enriched, compared to non-RNase treated ERGO-1 IPs (**Figure 3G and H**). Collectively, these data suggest that the miRNA enrichment present in the ERGO-1 IPs may be indirect due to an interaction between ERGO-1 and ALG-1 or ALG-2 on target transcripts. They also imply that co-regulation of target transcripts by 26G-RNAs and miRNAs could occur. Finally, these observations highlight an important consideration for interpreting IP/sRNA sequencing data: the association of multiple AGOs on common transcripts could result in skewed sRNA enrichment patterns, thus additional experiments such as those described above and examination of *ago* mutant sRNA sequencing data are warranted.

To explore the functional and developmental consequences of physical interaction between ERGO-1 and ALG-1, we created *alg-1; ergo-1* double mutants. Loss of *alg-1* results in heterochronic phenotypes; however, loss of *ergo-1* has no obvious phenotypic impact, aside from enhanced ability to perform exogenous RNAi (Eri phenotype). *alg-1; ergo-1* double mutants appeared sickly, being smaller and more pale than wild-type animals, with many dying prematurely, and only ~4% of

the animals surviving past the L4 stage, compared to ~45% and 100% for *alg-1* and *ergo-1* single mutants, respectively (**Figure 3I**). These data indicate that *alg-1* and *ergo-1* genetically interact to ensure survival into adulthood, and are consistent with the idea that coordinated regulation of targets by these AGOs is required for development.

## Novel miRNAs associated with ALG-1/2

The sequencing depth of our IP experiments allowed us to identify novel and lowly expressed miRNAs that may have previously eluded detection, been mis-annotated or otherwise not appreciated as bona fide miRNAs because they are not known to associate with a classical (miRNA) AGO. We used the miRNA prediction program mirDeep2 to analyze sequencing data from the miRNA binding AGO IPs (**Friedländer et al., 2008**). We found 10 putative, high-confidence miRNAs (**Supplementary file 5**) present in ALG-1 and ALG-2 complexes. These putative miRNAs are present at very low levels, <1 RPM on average in total sRNA samples (for comparison, the abundant miRNA let-7 is present at ~6000 RPM levels) (**Figure 3J**). Consistent with these being bona fide miRNAs, 5 of the 10 putative miRNAs were significantly depleted in *alg-1* or *alg-2* mutants (**Figure 3K**). Collectively, our results indicate these sRNAs are genuine miRNAs as they bind to classical miRNA AGOs and rely on these AGOs for their stability (**Figure 3L**).

# piRNAs

## 'miRNAs' associated with PRG-1 are mis-annotated piRNAs

We found three likely mis-annotated miRNAs (cel-miR-4936, cel-miR-8198-3p, and cel-miR-8202-5p) enriched in IPs of the Piwi AGO, PRG-1, consistent with a previous report of two similarly mis-annotated miRNAs (cel-miR-78 and cel-miR-798) (**Batista et al., 2008**). These miRNAs are 21 nt long and have a 5' uridine (**Figure 4A**); however, four of five were not enriched in association with the miRNA binding AGOs (cel-miR-4936, cel-miR-798, cel-miR-8198-3p, and cel-miR-8202-5p). All five miRNAs were depleted over 60-fold on average in *prg-1* mutants, were only twofold depleted on average in *rde-1* mutants, and were not depleted in canonical miRNA binding *ago* mutants *alg-1*, *alg-2*, and *alg-5* (**Figure 3C**, **Supplementary file 4**). These putative piRNA loci possess upstream regulatory sequences that fully or partially resemble Ruby motifs, found at most piRNA loci (**Ruby et al., 2006**; **Figure 4A**). Thus, these five sRNAs are piRNAs that were mis-annotated as miRNAs, demonstrating the utility of AGO IP-sequencing data in annotating sRNA features in the genome.

## piRNAs associate with the Argonaute PRG-1

Across all sRNA datasets, we detected reads for 14,568 out of the 15,363 annotated piRNAs in the genome. Among all IP data sets, we found 5943 piRNAs enriched over twofold, with Piwi PRG-1 being associated with nearly every enriched piRNA (5932/5943). The levels of piRNAs in *ago* mutants revealed that *prg-1* is required to maintain piRNA pools globally, with an ~13.9-fold reduction in piRNA levels overall (**Figure 4B**), consistent with previous observations (**Batista et al., 2008**; **Das et al., 2008**; **Wang and Reinke, 2008**).

We also observed that sRNAs from annotated genomic features other than piRNAs were enriched in the PRG-1 IPs. Most of these reads were 21 nucleotides long, possessed a 5' uridine, and were depleted in *prg-1* mutants (**Figure 4C**). In some instances, 21U sRNA sequences partially overlapped annotated piRNAs; for example, a 21U sRNA sequence was shifted by 3 nucleotides from the 5' end of the piRNA *21ur-2789* (**Figure 4D**). In total, we found 466 sequences that were 21U, enriched over twofold in both PRG-1 IP replicates, and were not perfectly matching to annotated piRNA sequences. All of these 21U sRNAs were depleted, and 222 were significantly depleted, in *prg-1* mutants compared to wild-type (**Figure 4E**, **Supplementary file 6**). Sequence logo analysis of the sequence upstream of the loci generating 21U sRNAs demonstrates similarity to the Ruby motifs of piRNAs (**Figure 4F**), and most of these sequences (375/466) originated from a previously described piRNA cluster that spanned coordinates 4.5–7.0M on chromosome IV (**Ruby et al., 2006**; **Figure 4G**). We conclude that these 466 21U sRNAs are previously uncharacterized piRNAs (**Figure 4H**).

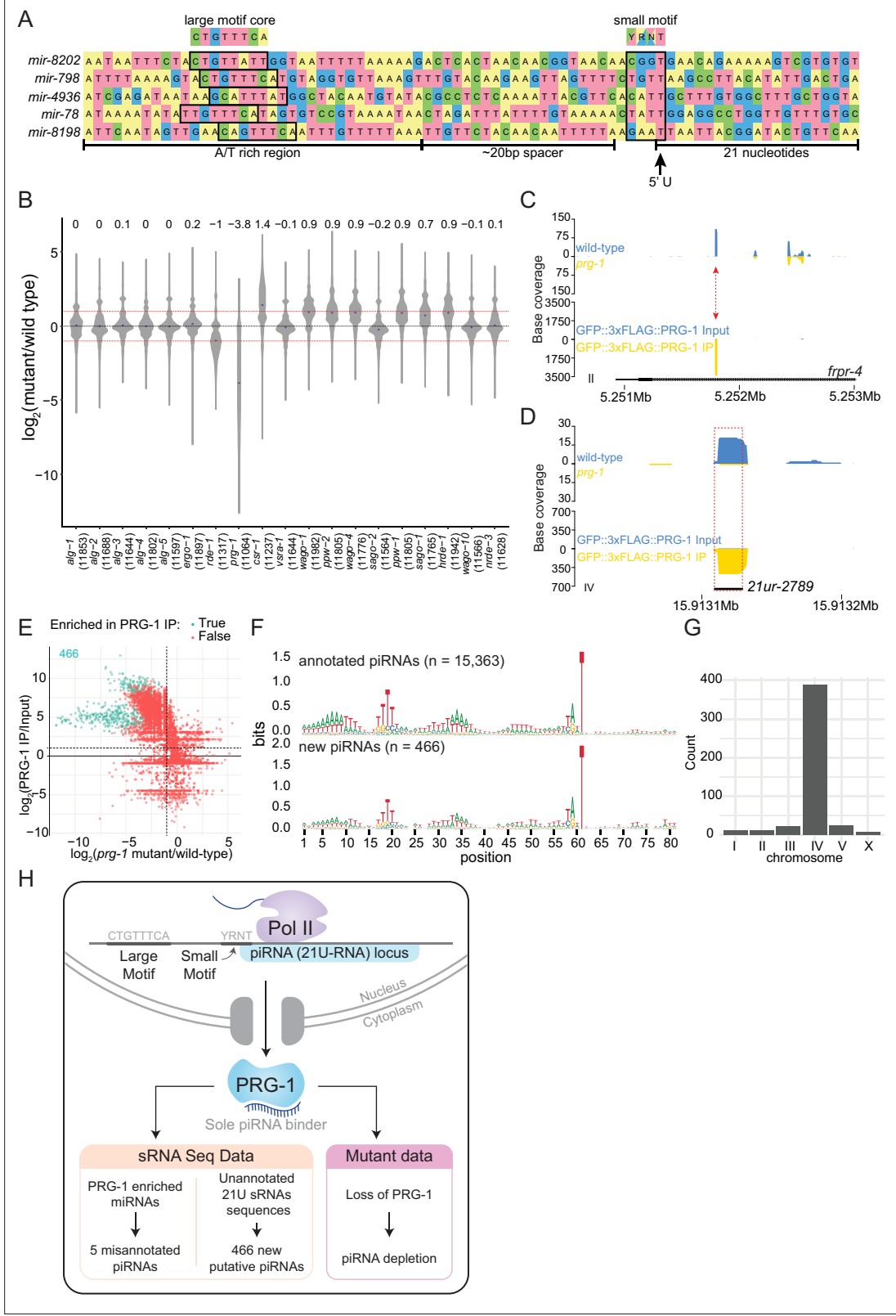

**Figure 4.** Analysis of piRNAs in AGO IPs and *ago* mutants reveals new piRNAs. (**A**) Genomic loci of annotated miRNAs that are enriched in PRG-1 IPs and suspected to be piRNAs. Ruby motifs highlighted above. (**B**) Violin plots depicting the fold change of all detected piRNAs in *ago* mutants compared to wild-type. (**C**) An example of a novel piRNA sequence, where a 21U sRNA sequence that was enriched in PRG-1 IPs and depleted in *prg-1* mutants originated from the intron of the gene *frpr-4* in the antisense orientation (red arrow). Note the difference in scales. (**D**) An example of a

*Figure 4 continued on next page*

Figure 4 continued

novel piRNA sequence originating from a shift of 3 nt from the annotated *21ur-2789* piRNA (dotted red box, black line). Note the difference in scales. (**E**) Scatter plot showing individual expression levels of 21U sRNAs that are not annotated as piRNAs. The y-axis shows enrichment in PRG-1 IPs and the x-axis shows depletion in *prg-1* mutants. The cyan dots represent individual 21U sRNAs which are twofold enriched in PRG-1 IPs. (**F**) Sequence logo analysis of annotated piRNA loci (top) and the new 466 piRNA loci (bottom). (**G**) Chromosome distribution of the 466 putative piRNA sequences. (**H**) A summary of piRNA pathway observations.

## Endo-siRNAs: 22G-RNAs and 26-RNAs

### Endo-siRNA-associated AGOs cluster into four groups

We examined the AGOs that interact with 22G- and 26G-RNAs. Because these sRNAs are generated by RdRPs, we can predict their targets based on sequence complementarity. We focused on endo-siRNAs antisense to protein-coding genes, pseudogenes, lincRNAs, and repetitive and transposable elements as these are the most abundant targets of the endo-siRNA binding AGOs (*Supplementary file 2*). For this analysis, we defined 22G-RNAs as reads that are 20–24 nt long and 26G-RNAs as reads that are 25–27 nt long with no 5′ nucleotide restriction for either sRNA species (*Supplementary file 3*).

Of the 19,999 annotated protein-coding genes, we detected endo-siRNA reads against 19,579 genes across all data sets, suggesting that sRNAs are generated against the entire protein-coding transcriptome at some level. A total of 10,127 genes had sRNAs that were enriched at least twofold in association with at least one AGO, indicating that at least half of the protein-coding genome has the potential to be regulated by AGO/sRNA complexes at this developmental stage. Hierarchical clustering of the enriched target genes of each AGO enabled us to identify four clear clusters of AGOs that target similar sets of protein-coding genes and could therefore function together in regulating those genes (*Figure 5A and B*). We compared the gene targets of each AGO to previously described sRNA pathways: (1) ALG-3 and -4 26G-RNA targets, defined as targets that are depleted of sRNAs in *alg-3; alg-4* mutants (1428 targets, *Conine et al., 2010*); (2) ERGO-1 26G-RNA targets, defined as enriched in ERGO-1 IPs (60 targets, *Vasale et al., 2010*); (3) CSR-1 targets, defined as enriched in CSR-1 IPs (4230 targets, *Claycomb et al., 2009*); (4) WAGO targets, defined as the overlap of sRNAs depleted in *rde-3*, *mut-7*, and *mago12* (a strain containing null mutations in 12 *wagos*) (1136 targets, *Gu et al., 2009*); and (5) Mutator targets defined as targets that are depleted of sRNAs in *mut-16* mutants (3625 targets, *Phillips et al., 2012*). We compared the gene targets of each AGO to the genes depleted of sRNAs in the RdRP mutants *rrf-1* (131 targets), *rrf-3* (319 targets), *ego-1* (5403 targets), and *ego-1; rrf-1* (6595 targets, *Sapetschnig et al., 2015*) to determine which RdRP generates each type of AGO-associated sRNA. We also compared the AGO-enriched targets to sRNA targets that are enriched (82 transcripts) or depleted (4357 transcripts) of sRNAs in germline-less *glp-4(bn2)* mutants (*Gu et al., 2009*), representing sRNAs that are enriched in the soma or germline, respectively.

The first AGO cluster consists of the WAGOs WAGO-1 (1814 targets), PPW-2 (636 targets), HRDE-1 (1295 targets), and PPW-1 (870 targets) (*Figure 5A*), which we term the WAGO cluster. These AGO complexes are enriched for sRNAs targeting WAGO and Mutator class genes, and are largely depleted of sRNAs targeting ALG-3 and -4, ERGO-1, and CSR-1 class genes (*Figure 5C*). The targets of these WAGOs also strongly overlap with targets of sRNAs depleted in *ego-1; rrf-1* double mutants (*Figure 5C*). Moreover, the targets of these WAGOs primarily overlap with transcripts depleted of sRNAs in *glp-4* mutants (*Figure 5C*). These observations are consistent with the germline expression of WAGO cluster AGOs (*Figure 6A*). Gene Ontology (GO) analysis of the WAGO cluster targets shows enrichment for kinase activity and protein binding, along with signaling, motility, and morphogenesis (*Supplementary file 6*).

The second AGO cluster consists of the AGOs SAGO-2 (1537 targets), SAGO-1 (181 targets), ERGO-1 (239 targets), and NRDE-3 (116 targets) (*Figure 5A*), which we term the ERGO-1 cluster. These AGOs are enriched for previously described ERGO-1 and Mutator target genes and depleted for ALG-3 and -4, CSR-1, and WAGO targets (*Figure 5C*). By comparing the ERGO-1 cluster targets to genes depleted of sRNAs in RdRP mutants, we observed overlap with *rrf-3* and *rrf-1* but not *ego-1* (*Figure 5C*). The ERGO-1 cluster also overlaps with *glp-4* sRNA-enriched transcripts (somatic genes) and is largely depleted of *glp-4* sRNA-depleted transcripts (germline genes) (*Figure 5C*). Consistent with the expression of ERGO-1, these data indicate that the targets of the ERGO-1 cluster are largely somatic. GO analysis of the targets of the ERGO-1 cluster revealed enrichment in various

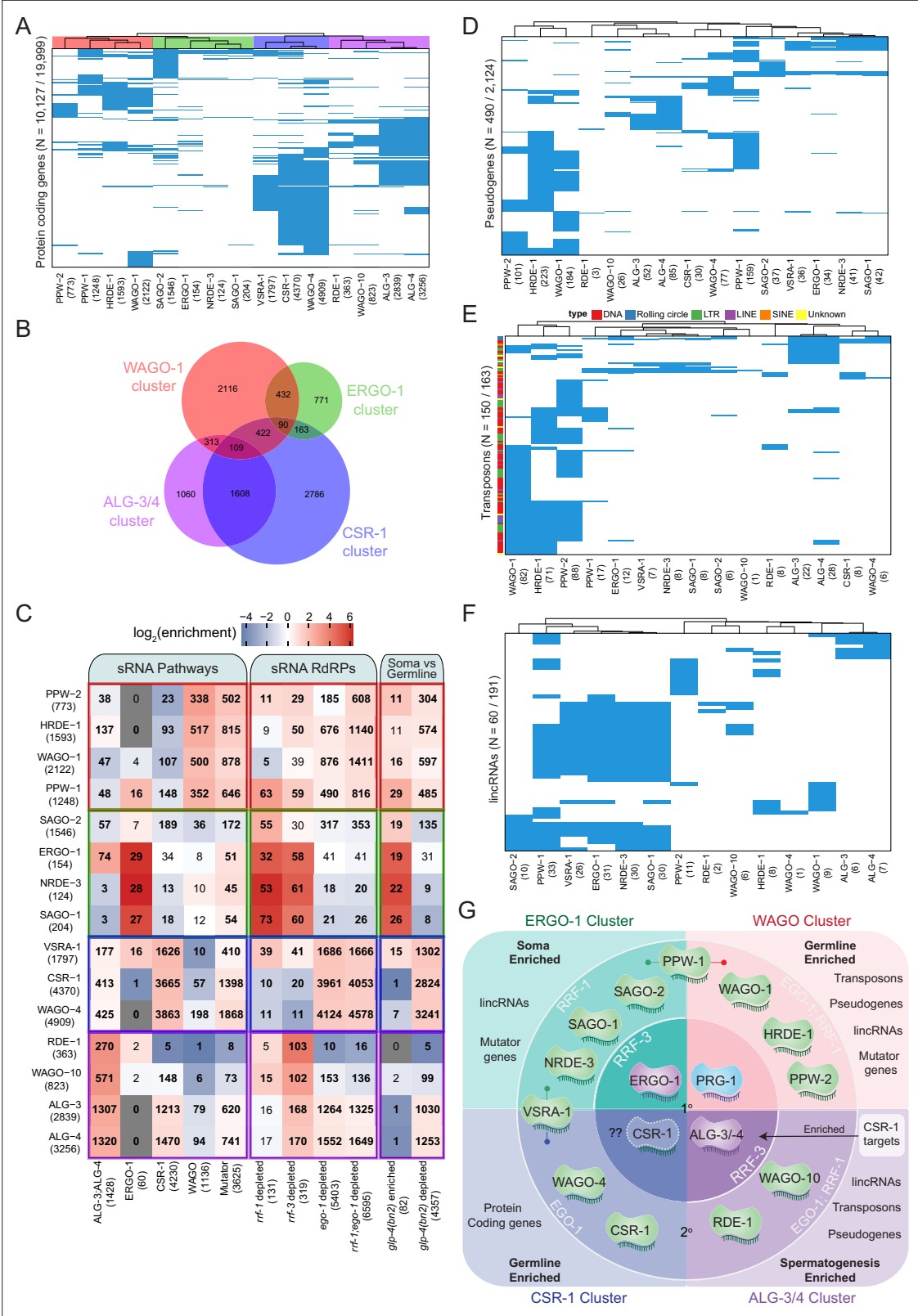

**Figure 5.** Analysis of endo-siRNA binding AGOs reveals functional categorization of AGOs to regulate distinct genetic elements. (**A**) Clustering diagram of AGO protein-coding gene targets. Each blue line represents a transcript for which endo-sRNAs were enriched twofold or more relative to input in both IPs. 22G-RNAs (defined as 20–24 nt with no 5′ nucleotide bias) were considered for all AGOs except ALG-3, ALG-4, and ERGO-1, for which only 26G-RNAs (defined as 25–27 nt with no 5′ nucleotide bias) were considered. (**B**) A Venn diagram showing the overlaps of protein–coding gene targets

*Figure 5 continued on next page*

*Figure 5 continued*

of the AGO clusters as highlighted by the color scheme in (**A**). (**C**) Enrichment analysis of the AGO protein–coding gene targets in previously described datasets. (**D**) Clustering diagram of AGO pseudogene targets as in (**A**). (**E**) Clustering diagram of AGO transposon targets as in (**A**). (**F**) Clustering diagram of AGO lincRNA targets as in (**A**). (**G**) A schematic summary highlighting the major AGO/sRNAs networks uncovered by endo-siRNA analysis.

The online version of this article includes the following figure supplement(s) for figure 5:

**Figure supplement 1.** Metagene analysis of sRNAs.

**Figure supplement 2.** Additional analysis of the endo-siRNA binding AGO IPs.

**Figure supplement 3.** Loss of *ago* genes results in differential effects of sRNA levels associated with each AGO.

developmental processes, and the most significant terms were those associated with immune, defense, and stress responses (*Supplementary file 6*).

The third cluster consists of the WAGOs CSR-1 (4182 targets), WAGO-4 (4815 targets), and C04F12.1/VSRA-1 (1797 targets) (*Figure 5A*), which we term the CSR-1 cluster. The targets of this cluster significantly overlap with the set of previously described CSR-1 targets and are depleted for WAGO targets (*Figure 5B and C*). This cluster also significantly overlaps with the 26G-RNA targets of ALG-3 and -4 and Mutator targets (*Figure 5B and C*), consistent with recent observations detailing the sRNA association of each CSR-1 isoform (*Charlesworth et al., 2021*; *Nguyen and Phillips, 2021*). It is important to keep in mind that the CSR-1 strain we used tags both isoforms of CSR-1, and these IPs were performed at a developmental time when CSR-1b is highly expressed in the oogenic germline while CSR-1a is expressed at lower levels in the intestine. The CSR-1 cluster significantly overlapped with genes depleted of sRNAs in *ego-1* and in *glp-4* mutants, as previously described (*Claycomb et al., 2009*), and consistent with germline expression (mostly attributable to CSR-1b). The CSR-1 cluster targets are associated with many biological process GO terms (up to 604, *Supplementary file 7*), including terms related to meiosis and chromosome segregation for which CSR-1b is known to be essential (*Claycomb et al., 2009*; *Charlesworth et al., 2021*; *Nguyen and Phillips, 2021*).

The fourth cluster consists of the AGOs RDE-1 (388 targets), WAGO-10 (877 targets), ALG-3 (2966 targets), and ALG-4 (3378 targets) (*Figure 5A*), which we term the ALG-3/4 cluster. This cluster is enriched for previously described targets of ALG-3 and ALG-4 26G-RNAs and depleted of WAGO targets (*Figure 5B and C*). The cluster can be further subdivided into ALG-3 and -4 versus WAGO-10 and RDE-1, where ALG-3 and -4 are also enriched for CSR-1 and Mutator targets and WAGO-10 and RDE-1 are depleted for such targets (*Figure 5C*). These AGOs are also depleted of previously published ERGO-1 26G-RNA targets (*Figure 5C*). The ALG-3/4 cluster AGOs significantly overlapped with genes depleted of sRNAs in *rrf-3* mutants and in *ego-1* mutants (*Figure 5C*). The ALG-3/4 cluster showed significant overlap with genes depleted of sRNAs in *glp-4* mutants, consistent with targeting germline-enriched genes (*Figure 5C*). GO term analysis reveals many biological processes are regulated by the ALG-3/4 cluster, including terms associated with gamete generation, and specifically spermatogenesis (*Supplementary file 7*), consistent with previous roles attributed for ALG-3 and ALG-4 (*Conine et al., 2010*), and with the spermatogenic-restricted expression patterns of ALG-3, ALG-4 and WAGO-10 (*Charlesworth et al., 2021*; *Figure 6A*).

## sRNA profiles suggest similar sRNA biogenesis and targeting mechanisms

The distribution of sRNAs along sets of target transcripts may provide insights into the mechanisms of sRNA biogenesis and target regulation. To visualize sRNA distribution across sets of protein-coding targets for each AGO, we used metagene plots (*Figure 5—figure supplement 1*). Patterns of sRNA distribution were generally more similar within AGO clusters than between clusters; however, there were some differences. The WAGO cluster generally displays high levels of sRNA enrichment, and these sRNAs are present over most of the transcript (*Figure 5—figure supplement 1*). The ERGO-1 cluster has the highest levels of sRNAs overall, which tend to be either distributed over most of the target transcript (SAGO-1 and NRDE-3) or biased toward the 3′ end (SAGO-2 and ERGO-1) (*Figure 5—figure supplement 1*). The CSR-1 cluster shows strong similarity between the WAGO-4 and CSR-1 sRNA distribution profiles, with enrichment at the 3′ and to a lesser extent the 5′ end (*Charlesworth et al., 2021*), while the profile of C04F12.1/VSRA-1 sRNAs appears more similar to that of SAGO-1 and NRDE-3 (*Figure 5—figure supplement 1*). Consistent with previous observations, the 26G-RNAs associated with the ALG-3/-4 pathway are enriched at the 5′ and 3′ ends of target transcripts (*Conine*

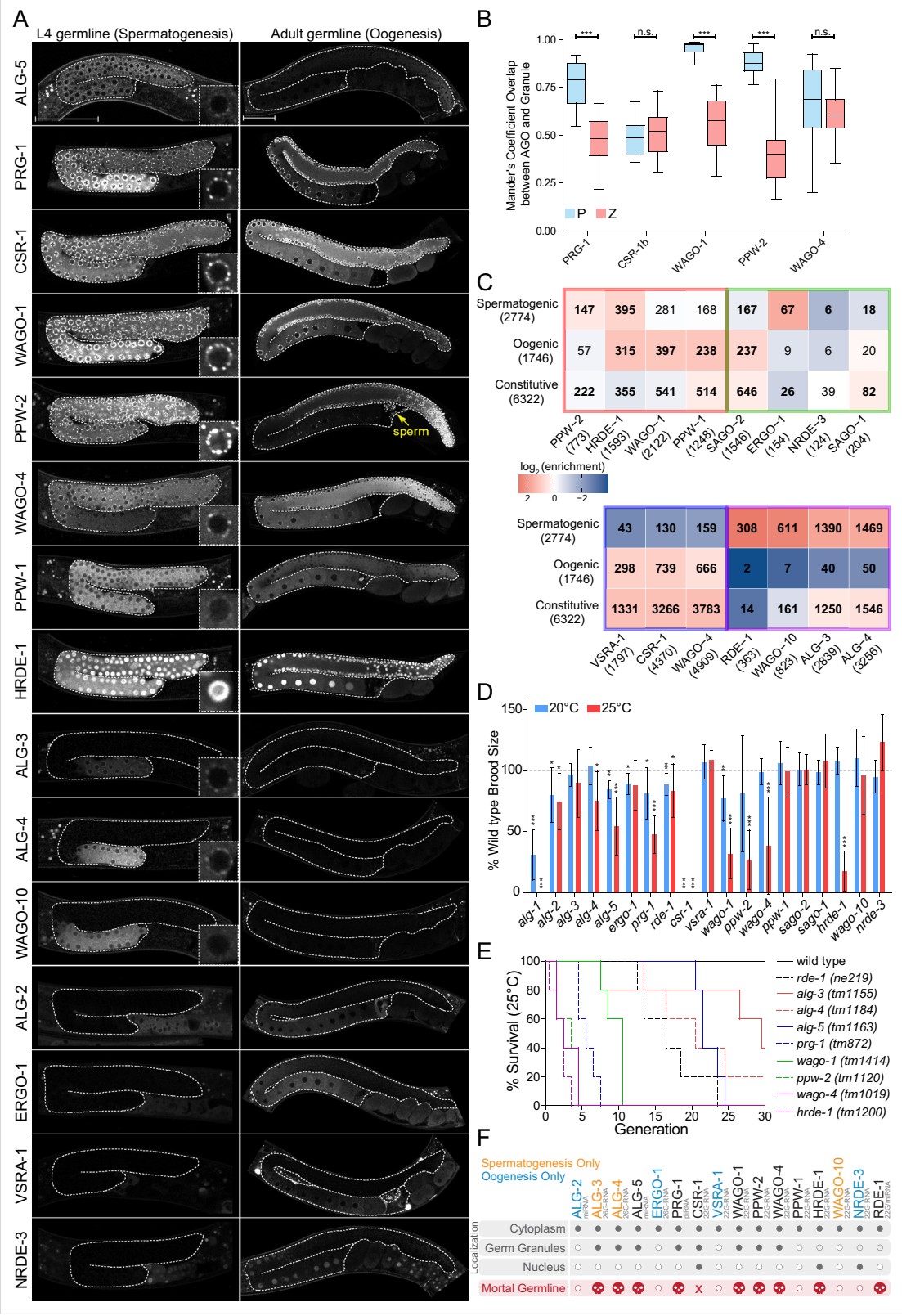

**Figure 6.** AGOs are differentially expressed in the germline and differentially regulate germline gene expression to promote fertility. (**A**) Expression patterns of germline AGOs in L4 (left) and adult (right) germlines. Inset image zoomed in at an individual germ cell nucleus. Yellow arrow points to sperm within the spermatheca. Scale bar represents 50µm. Due to low levels of expression, RDE-1 is not shown. See *Figure 7—figure supplement 8* for expression of the *rde-1p::gfp* transcriptional reporter. (**B**) Quantification of the number of GFP::AGO pixels that overlap with PGL-1::mRFP (blue) or

*Figure 6 continued on next page*

*Figure 6 continued*

HA::TagRFP::ZNFX-1 (coral) pixels using Mander's correlation. For each data set, five Z stacks of proximal germline regions from six different animals per strain were counted (N = 30 slices, approximately 80–100 nuclei per worm). ***p-value<0.001, n.s. = not significant. One-way ANOVA with Bonferroni's post hoc multiple comparison test. (**C**) Analysis of the enriched sRNA targets in each of the AGOs in comparison to germline constitutive, oogenic, and spermatogenic expressed genes (*Ortiz et al., 2014*). Bold numbers indicate significant enrichment or depletion (p<0.05), Fisher's exact test. The colored borders represent the AGO clusters as defined in *Figure 5A*. (**D**) Brood size analysis of all ago mutants at 20°C and 25°C. Data was aggregated from different experiments and normalized to the mean of wild-type control samples. *p<0.05, **p<0.01, ***p<0.001, two-sided t-test. N ≧ 10 P0 worms. Error bars represent standard deviation. (**E**) Mortal germline assay of *ago* mutants showing a Mrt phenotype in (**D**). N = 5 P0 worms. (**F**) A summary of the spatial and temporal localization of AGOs in the germline and Mrt phenotypes. CSR-1 has an 'X' to indicate it is essential. AGOs in black are expressed in both spermatogenesis and oogenesis.

The online version of this article includes the following figure supplement(s) for figure 6:

**Figure supplement 1.** Additional analysis of AGO mRNA expression.

**Figure supplement 2.** The mortal germline of *wago* mutants is reversible and associated with reduced germline proliferation.

---

*et al., 2010*; *Figure 5—figure supplement 1*). We observed a similar 5′ enrichment for the 22G-RNAs associated with RDE-1 and WAGO-10, but not a 3′ sRNA enrichment (*Figure 5—figure supplement 1*). These observations may be indicative of differences in sRNA biogenesis or targeting mechanisms, and will necessitate further study.

## Rapidly evolving and 'non-self' elements in the genome are targeted by distinct AGO/sRNA pathways

Although protein-coding genes comprise the majority of sRNA targets, we also examined pseudogenes, repetitive and transposable elements, and lincRNAs. We detected sRNA reads against 1741 out of 2124 annotated pseudogenes and found that 490 were enriched with antisense sRNAs across our IP data sets. Hierarchical clustering of the enriched targets of each AGO largely resembled the four clusters observed for the protein-coding targets of the AGOs (*Figure 5D*) with two exceptions: PPW-1 and C04F12.1/VSRA-1 clustered with the ERGO-1 pathway instead of the WAGO-1 pathway (PPW-1) and CSR-1 pathway (C04F12.1/VSRA-1). The majority of pseudogenes were targeted by the WAGO cluster (*Figure 5D*, *Figure 5—figure supplement 2A*). Many pseudogenes were specifically enriched in one pathway over the others. For example, out of the 95 pseudogenes targeted by the ALG-3/4 cluster, 59 were unique to this cluster (*Figure 5—figure supplement 2A*). These differences may reflect tissue-specific regulation by different pathways or partitioning of transcripts into separate pathways.

We detected reads against all 163 annotated transposable elements. Of these, 150 were enriched for antisense sRNAs across our IP data sets (*Figure 5E*). This indicates that all transposable elements have the potential to be regulated by sRNAs and that most of them are likely to be highly regulated by AGO/sRNA complexes. As with pseudogenes, the PPW-1 and C04F12.1/VSRA-1 targets overlapped with the ERGO-1 cluster targets, but this cluster appears to regulate a relatively small number of transposable elements overall, with only one TE uniquely regulated by this cluster (*Figure 5E*, *Figure 5—figure supplement 2B*). ALG-3/-4 regulate a group of 36 transposable elements that are likely to be expressed during spermatogenesis. Most transposable elements (134) are targeted by the WAGO cluster AGOs, with 104 of these being unique to this cluster (*Figure 5E*, *Figure 5—figure supplement 2B*). This suggests that the WAGO cluster is the major regulator of transposable elements, particularly in the germline.

Transposable elements can be subdivided based on their class and method of transposition: DNA transposons (cut-and-paste DNA transposons, Rolling circle replication DNA transposons), and retrotransposons (LTR, LINE, SINE). We observed that different AGOs show differential sRNA enrichment for specific types of transposons. For example, the ERGO-1 cluster AGOs are relatively depleted of cut-and-paste DNA transposon, LINE and LTR element-targeting sRNAs but enriched for Rolling circle-targeting sRNAs (*Figure 5—figure supplement 2C*). In contrast, the WAGO cluster was generally enriched for sRNAs targeting all other elements except Rolling circle transposons (*Figure 5—figure supplement 2C*).

We detected reads against 168 of the 191 annotated lincRNAs (defined in *Nam and Bartel, 2012* and WormBase version WS276), across all IP data sets, and, among these, 60 of these were enriched for antisense sRNAs (*Figure 5F*). Again, PPW-1 and C04F12.1/VSRA-1 lincRNA targets overlapped

with the ERGO-1 cluster targets (*Figure 5F*). The majority of lincRNAs (48) are targeted by the WAGO cluster, followed by the ERGO-1 cluster (35). Many of the targets overlapped between the WAGO and ERGO-1 clusters (26), with the majority of this overlap (25/26) originating from PPW-1 targets (*Figure 5—figure supplement 2D*). Seventeen lincRNA targets were specific to the WAGO cluster, three were specific to the ALG-3/4 cluster, and three were specific to the ERGO-1 cluster (*Figure 5—figure supplement 2D*). Our results point to lincRNAs as another important, yet poorly studied category of sRNA targets with the potential for tissue-specific regulation.

## Hierarchy and interconnected regulation of multiple sRNA pathways

To begin to understand why *C. elegans* encodes so many AGOs, we must first understand how the different sRNA pathways interact with each other. Loss of one component may affect the sRNA landscape in the organism, potentially allowing us to infer the hierarchical relationship between sRNA/AGO pathways. Our analysis indicated that some AGOs may have distinct roles and participate in different pathways depending on the tissue in which they are expressed. To address how loss of one component may affect the system, we sequenced sRNAs from all *ago* mutants in triplicate. We examined how the sRNAs antisense to protein-coding, pseudogene, repetitive and transposable element, and lincRNA targets of each AGO are affected in each *ago* mutant (*Figure 5—figure supplement 3*).

It is possible that AGOs may regulate other AGOs, especially as among the targets of AGO/sRNAs were other *agos* (*Figure 5—figure supplement 2E*), indicating there may be regulatory networks in place in which sRNA pathways can regulate others, and/or participate in regulatory feedback loops. The WAGOs CSR-1 and WAGO-4 form a hub that targets most of the constitutively expressed germline AGOs (see below; note that this is mostly attributable to CSR-1b based on the developmental time assayed), while the spermatogenesis-specific AGOs, ALG-3, ALG-4, and WAGO-10 form a self-regulatory hub (*Figure 5—figure supplement 2E*). Below we outline several observations from this analysis and put these results in the context of our current knowledge of hierarchy in sRNA targeting.

Loss of the miRNA-binding AGOs ALG-1, ALG-2, and ALG-5 did not have large effects on the endo-siRNA pathways (*Figure 5—figure supplement 3*); therefore, we begin our analysis by focusing on established primary AGOs: PRG-1, ERGO-1, and ALG-3/4. Given that CSR-1 seems able to play both primary and secondary AGO roles, we include it in this group. Loss of the piRNA-binding primary AGO PRG-1, which results in the loss of most piRNAs (*Figure 4B*), led to downregulation of secondary sRNAs associated with the WAGO cluster AGOs sRNAs (*Montgomery et al., 2021*). This result is consistent with the model that piRNA targeting induces the production of 22G-RNAs that are then loaded into WAGO-1 and HRDE-1 (*Lee et al., 2012*; *Cornes et al., 2022*; *Das et al., 2008*; *Bagijn et al., 2012*). Our complete analysis reveals that PPW-1 and PPW-2 also work within the context of PRG-1 targeting. With similar logic, loss of the 26G-RNA-binding primary AGO ERGO-1 resulted in the loss of ERGO-1 cluster sRNAs (*Figure 5—figure supplement 3*). Consistent with our observation that NRDE-3 sRNAs are depleted, it was previously shown that the translocation of NRDE-3 to the nucleus requires sRNA binding, and loss of ERGO-1 results in NRDE-3 remaining cytoplasmic (*Guang et al., 2008*). Our complete analysis reveals that SAGO-1- and SAGO-2-associated 22G-RNAs are depleted upon loss of ERGO-1, indicating that SAGO-1 and SAGO-2 also work within the context of ERGO-1 targeting.

We observed that the sRNAs of some 22G-RNA-associated AGOs are differentially affected by loss of primary AGOs. For example, PPW-1-bound sRNAs that target protein-coding genes, pseudogenes, and transposons are depleted in *prg-1* mutants but not *ergo-1* mutants (*Figure 5—figure supplement 3*). Conversely, PPW-1-bound sRNAs that target lincRNAs are depleted in *ergo-1* mutants but not *prg-1* mutants (*Figure 5—figure supplement 3*). When taken together with the clustering analysis of targets (*Figure 5A and D–F*), it appears that PPW-1 acts downstream of both the PRG-1 piRNA pathway and the ERGO-1 26G-RNA pathway. This PPW-1 duality may be dependent on the tissue (germline and soma) in which it is expressed. Thus, it is likely that in the germline PPW-1 acts downstream of PRG-1 piRNAs, and in the soma it acts downstream of ERGO-1 26G-RNAs. A similar observation is made for C04F12.1/VSRA-1-bound sRNAs. C04F12.1/VSRA-1 acts downstream of ERGO-1 26G-RNAs to target pseudogenes, transposons, and lincRNAs, and acts in conjunction with CSR-1 22G-RNAs to target protein-coding genes in the germline (*Figure 5A and D–F*, *Figure 5—figure supplement 3*). Because of this dual association with the CSR-1 cluster and the ERGO-1 cluster, we named C04F12.1 VSRA-1, for <u>V</u>ersatile <u>S</u>mall <u>R</u>NAs <u>A</u>GO.

Loss of the 26G-RNA-binding AGOs ALG-3 and ALG-4 individually did not affect sRNA populations, likely due to their partial redundancy with each other (*Figure 1E*, *Conine et al., 2010*). Interpreting the loss of CSR-1 is more complicated, given that it targets nearly one quarter of the protein-coding genome, is encoded as two isoforms with distinct expression profiles and functions, and is required for both gene licensing as well as silencing (*Charlesworth et al., 2021*; *Nguyen and Phillips, 2021*). Loss of both isoforms of CSR-1 in a null mutant resulted in depletion of CSR-1 cluster sRNAs targeting protein-coding genes (*Figure 5—figure supplement 3*); however, the sRNA levels of other AGOs, including WAGO-1, HRDE-1, PPW-1, SAGO-2, and SAGO-1, were also decreased. This may be due to secondary effects arising from loss of CSR-1, which appears to be capable of regulating many other AGOs (*Figure 5—figure supplement 2E*).

Loss of single 22G-RNA-associated AGOs had different effects depending on the AGO (*Figure 5—figure supplement 3*). Like loss of both CSR-1 isoforms, loss of WAGO-4 also resulted in depletion of CSR-1 cluster sRNAs. Loss of the WAGO cluster WAGOs had varying effects. Loss of WAGO-1 or HRDE-1 resulted in the depletion of PPW-2-associated sRNAs, while loss of HRDE-1 alone also resulted in depletion of WAGO-1- and HRDE-1-associated sRNAs. This indicates that HRDE-1 is required for the stability of most of the WAGO cluster associated sRNAs, likely reflecting the requirement of HRDE-1 in producing tertiary sRNAs (*Sapetschnig et al., 2015*). Loss of SAGO-1 resulted in downregulation of ERGO-1 cluster-associated sRNAs primarily targeting pseudogenes and lincRNAs. This suggests SAGO-1 is the main AGO targeting these elements downstream of ERGO-1/26G-RNAs. Loss of PPW-2 did not result in downregulation of WAGO cluster small RNAs, but rather CSR-1 cluster sRNAs, highlighting the potential for interplay between pathways.

In summary, we have defined which classes of genetic elements are targeted by each AGO and the genetic requirements for each *ago* in accumulating sRNAs targeting these elements (*Figure 5G*). These results define the relationships between different AGO pathways and highlight the complexity of target regulation. The clear targeting of different genetic elements by different clusters of AGOs implies that features intrinsic to the target transcript likely encode determinants for AGO/sRNA pathway specificity. Furthermore, the spatiotemporal expression profile of targets, sRNA biogenesis components, and AGOs are also likely to be major contributors to the patterns we have observed.

## AGOs have distinct spatiotemporal localization patterns during development

To link our molecular observations with the AGO expression profiles and gain insight into where each AGO exerts its function, we visualized GFP fluorescence using confocal microscopy at each stage of worm development, revealing that AGOs are expressed in a variety of tissues throughout the life cycle. We identified 16/19 AGOs expressed in the germline, consistent with known roles for AGOs in the germline (*Figure 6A and F*, *Figure 6—figure supplement 1A*). Eight of these show germline-restricted expression and the other eight are also expressed in somatic tissues (*Figure 7F*, *Figure 7—figure supplements 1–8*). In total, 11/19 AGOs are expressed in the soma, with 3 AGOs being somatically restricted (SAGO-1, SAGO-2, ALG-1) (*Figure 7F*, *Figure 7—figure supplements 1–8*). We observed AGOs expressed in various somatic tissues, including but not limited to the vulva, hypodermis, muscle, seam cells, intestine, neurons, somatic gonad, and spermatheca (*Figure 7A and F Figure 7—figure supplements 1–8*). AGO expression levels vary, and low expression levels or expression under specific (e.g., stress induced) conditions may have precluded detection of some AGOs in some tissues. Specifically, RDE-1 was hardly detectable as a translational fusion. However, the first step in our CRISPR protocol generates a transcriptional reporter (*Dickinson et al., 2015*), by which we observed strong GFP expression under the control of the *rde-1* promoter. Thus, for RDE-1, we use the transcriptional reporter to deduce expression patterns (*Figure 7—figure supplement 8*).

## Germ granule-localized AGOs are required for transgenerational fertility

The majority of AGOs are expressed in the germline (16/19), where they display striking temporal and spatial localization patterns (*Figure 6A and F*). Nine AGOs are constitutively expressed throughout the germline from the emergence of the P1 cell through spermatogenesis in the L4 stage to oogenesis in adults: RDE-1, ALG-5, PRG-1, CSR-1b, WAGO-1, PPW-2, WAGO-4, PPW-1, and HRDE-1 (*Figure 6F*, *Figure 7—figure supplements 1–8*). Of these, two AGOs are more highly expressed

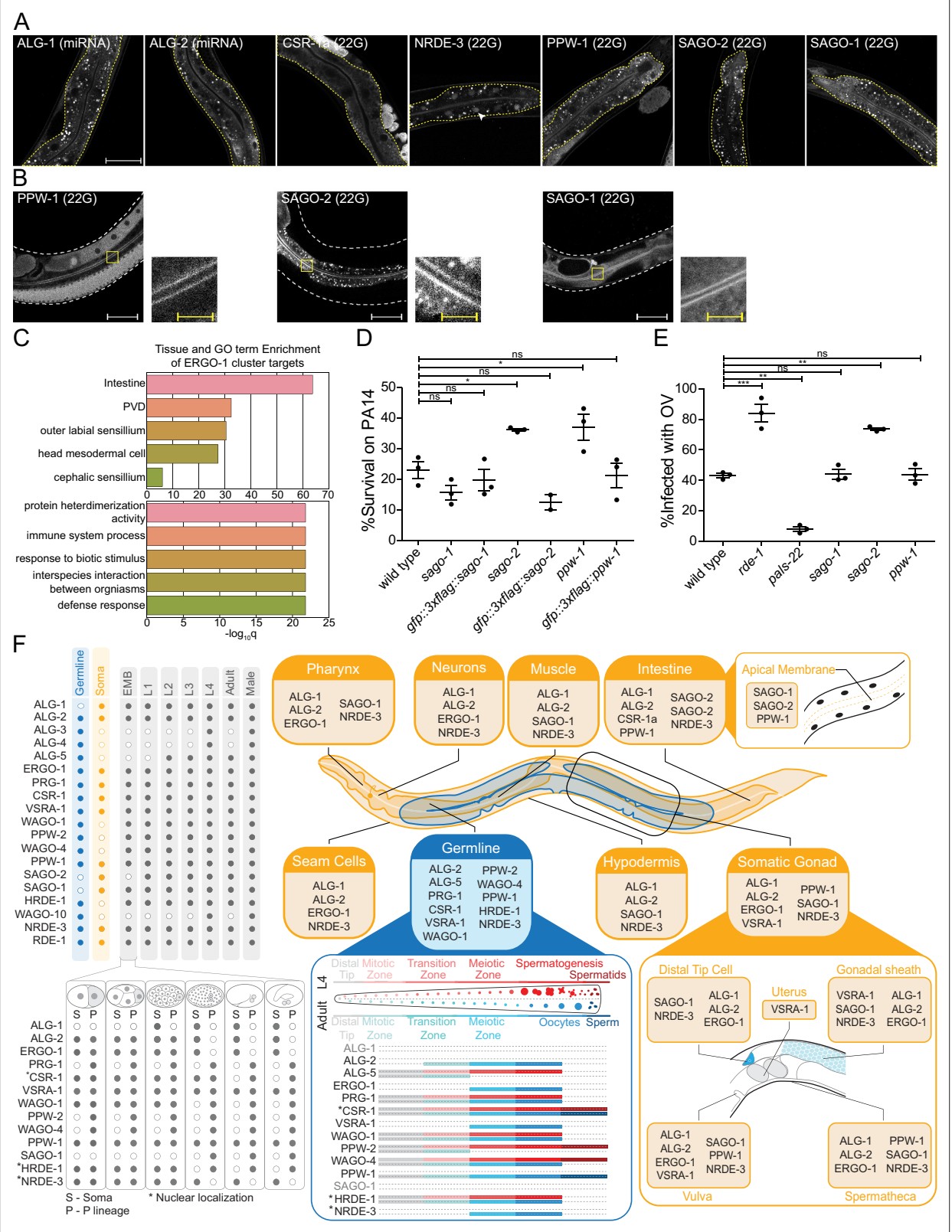

**Figure 7.** AGOs are expressed in multiple somatic tissues and several are required for normal immunity towards pathogens. (**A**) AGOs expressed in the intestine. Adult worms shown. Brackets indicate the type of sRNA AGOs associate with. Intestines are outlined in yellow. Arrowhead indicates intestine cell nuclei. Scale bar represents 50 μm for all images. Bright foci throughout intestinal tissue are autofluorescent gut granules. (**B**) Apical intestinal membrane localization of PPW-1, SAGO-2, and SAGO-1. Worm body is outlined in white. Zoomed-in panels are indicated with a yellow box. White

*Figure 7 continued*

scale bar represents 50 µm. Yellow scale bar represents 10 µm. Note that PPW-1 is also expressed in the germline. (**C**) Tissue enrichment analysis (top) and Gene Ontology analysis (bottom) of the ERGO-1 cluster sRNA targets. (**D**) Percent of worms alive after 72 hr of exposure to a *P. aeruginosa* PA14 lawn is shown for each strain. This is a representative experimental run out of three conducted. Each dot represents 50 worms. *p-value<0.05, n.s. = not significant. One-way ANOVA with Dunnett's post hoc multiple comparison test. (**E**) Percent of worms infected with Orsay virus (OV) 16 hr post infection is shown for each strain. Each dot represents ≥ 100 worms. N = 3. **p-value<0.01, ***p-value<0.001, n.s. = not significant. One-way ANOVA with Tukey post hoc multiple comparison test. (**F**) A summary of the expression patterns of all *C. elegans* AGOs throughout development.

The online version of this article includes the following figure supplement(s) for figure 7:

**Figure supplement 1.** AGO expression in embryos.

**Figure supplement 2.** AGO expression in L1 larvae.

**Figure supplement 3.** AGO expression in L2 larvae.

**Figure supplement 4.** AGO expression in L3 larvae.

**Figure supplement 5.** AGO expression in L4 larvae.

**Figure supplement 6.** AGO expression in adult worms.

**Figure supplement 7.** AGO expression in males.

**Figure supplement 8.** Representative images of developmental stages of the RDE-1 transcriptional reporter.

at the mitotic zone of the adult oogenic germline: ALG-5 and PPW-2 (*Figure 6A*). The other seven germline-expressed AGOs exhibit gamete-specific expression patterns. ALG-3, ALG-4, WAGO-10, and the CSR-1a isoform are expressed only during spermatogenesis, at the L4 stage in hermaphrodites and constitutively in the male germline (*Charlesworth et al., 2021*; *Nguyen and Phillips, 2021*; *Figure 6A*, *Figure 7F*, *Figure 7—figure supplement 7*). Only PPW-2 and CSR-1b are detectable in spermatids (*Figure 6A*, *Figure 7F*; *Charlesworth et al., 2021*; *Schreier et al., 2022*). ALG-2, ERGO-1, and NRDE-3 are specifically expressed in oocytes starting at the young adult hermaphrodite stage (*Figure 6A and F*). C04F12.1/VSRA-1 is also restricted to the oogenic germline in adult worms and shows expression in the primordial germ cells of L1 larvae (*Figure 6A*, *Figure 7—figure supplement 2*). Published mRNA expression data from dissected hermaphrodite and male germlines shows largely the same mRNA expression patterns as the tagged AGO proteins (*Figure 6—figure supplement 1B*; *Tzur et al., 2018*). All germline-expressed AGOs, except those that are expressed only during oogenesis, are also expressed in the male germline and display the similar patterns of expression (*Figure 7—figure supplement 7*).

The germline AGOs encompass different subcellular localization profiles. Half (8/16) of the germline AGOs are enriched in perinuclear phase-separated germ granules: ALG-3, ALG-4, ALG-5, PRG-1, CSR-1a, b, WAGO-1, PPW-2, and WAGO-4 (*Figure 6A and F*). PPW-1 is also weakly detected in germ granules, mostly restricted to the pachytene region of the adult germline (*Figure 6A*). There are four types of germ granules in *C. elegans* that are thought to play different roles in sRNA pathways: P granules, Z granules, Mutator Foci, and SIMR Foci (*Sundby et al., 2021*). PRG-1, CSR-1, and WAGO-1 were previously reported to co-localize with P granules (*Wang and Reinke, 2008*; *Batista et al., 2008*; *Claycomb et al., 2009*; *Gu et al., 2009*), and WAGO-4 was identified as a Z granule component (*Wan et al., 2018*). To determine which types of germ granules the germline-constitutive WAGOs, CSR-1b, and PRG-1 are associated with, we determined the overlap in pixels between fluorescently-tagged AGOs and components of P granules (PGL-1) or Z granules (ZNFX-1) using confocal microscopy in adult worms. The CSR-1 cluster AGOs CSR-1 and WAGO-4 overlapped roughly equally with both P and Z granule pixels, whereas the WAGO cluster AGOs PPW-2 and WAGO-1 were strongly biased toward overlap with P granules (*Figure 6B*). PRG-1 was intermediate to both of these phenotypes. Thus, AGOs with similar target preferences display similar germ granule localization patterns (*Figure 6B*).

Two AGOs are predominantly nuclear: HRDE-1 and NRDE-3 (*Figure 6A and F*). While HRDE-1 is known to be a nuclear germline AGO required for RNAi inheritance (*Buckley et al., 2012*), NRDE-3 was previously shown to be a somatic nuclear AGO, required for somatic RNAi inheritance (*Guang et al., 2008*). That we observed NRDE-3 localizing to oocyte nuclei may explain how it is capable of propagating certain RNAi responses into the next generation.

Due to the split in AGOs between germline-constitutive and sex-specific expression, we asked whether targets were also sex-biased. A previous report defined the spermatogenic, oogenic, and

germline-constitutive transcriptomes of *C. elegans* (*Ortiz et al., 2014*). Therefore, we compared the protein-coding target transcripts for 22G- and 26G-RNA binding AGOs to this dataset (*Figure 6C*). The 22G-RNA targets of the WAGO cluster AGOs PPW-2, HRDE-1, and WAGO-1 were depleted of germline constitutively expressed transcripts. Instead, PPW-2 and HRDE-1 targets were enriched for spermatogenic transcripts, and WAGO-1, HRDE-1, and PPW-1, targets were enriched for oogenic transcripts (*Figure 6C*). This suggests that the WAGO cluster can be further subdivided and is responsible for sex-specific gene regulation during the young adult stage when oogenesis occurs: PPW-2 regulates spermatogenic genes, and WAGO-1 and PPW-1 regulate oogenic genes. Both branches of this WAGO cluster appear to act upstream of or in parallel with the nuclear HRDE-1. The 22G-RNA targets of the CSR-1 cluster AGOs CSR-1, C04F12.1/VSRA-1, and WAGO-4 were depleted of spermatogenic transcripts and were enriched for both oogenic and germline-constitutive transcripts in the young adult stage (*Figure 6C*). We previously reported that CSR-1b and WAGO-4 associate with oogenic and germline constitutive transcripts during the L4 stage, and in contrast CSR-1a enriches for spermatogenic transcripts that overlap with the ALG-3/4 cluster (*Charlesworth et al., 2021*). During the L4 stage, targets of the ALG-3/4 cluster AGOs WAGO-10, ALG-3 and -4 were all enriched for spermatogenic transcripts and ALG-3 and -4 also showed enrichment for germline constitutive transcripts (*Figure 6C*). Similarly, during the young adult stage, the ALG-3/4 cluster AGO RDE-1 targets were also enriched for spermatogenic transcripts. Thus, germline AGOs have differential roles in regulating sex-specific germline transcripts in a spatiotemporal manner.

Given their germline expression, we next asked whether each germline AGO is required for fertility. Previous work examined a role for each *C. elegans ago* in fertility and found that only *csr-1* is essential at the normal laboratory-culturing temperature of 20°C (*Yigit et al., 2006*), although several other AGOs, including ALG-1, ALG-2, and PRG-1, have been shown to be required for optimal fertility at this temperature (*Batista et al., 2008*; *Bukhari et al., 2012*). However, in recent years it has become apparent that stressful conditions, such as elevated temperature, can have a substantial impact on the fertility of mutants. This temperature-dependent fertility defect can manifest in the first generation after a temperature shift or take several generations to reach its full impact. Several germ granule component mutants and *ago* mutants, including *hrde-1*, *prg-1*, *wago-4*, and *ppw-2*, have been shown to exhibit a Mortal germline (Mrt) phenotype, in which fertility decreases over several generations at elevated temperature (*Buckley et al., 2012*; *Simon et al., 2014*; *Schreier et al., 2022*; *Wan et al., 2018*). We therefore performed brood size assays at 20°C and 25°C for recently outcrossed hermaphrodites of each single *ago* mutant (*Figure 6D*). At 20°C, seven *ago* mutants showed compromised fertility: *alg-1*, *alg-2*, *alg-5*, *ergo-1*, *prg-1*, *rde-1*, and *wago-1* (*Figure 6D*). When shifted to 25°C, the same mutants displayed a low brood size, which was even more pronounced in some cases (e.g., *alg-1* was sterile). We also observed fertility defects not present at 20°C for four *ago* mutants: *alg-4*, *ppw-2*, *wago-4*, and *hrde-1* (*Figure 6D*). *csr-1* mutants were sterile at both temperatures (*Figure 6D*). These results indicate that more than half (10/19) of the *C. elegans* AGOs contribute individually to optimal fertility.

To further assess the role of temperature stress on the fertility of *ago* mutants, we conducted an Mrt assay, following worms for 30 generations at 25°C. We observed complete loss of fertility over varied numbers of generations for *alg-5* (25 generations), *rde-1* (24), *prg-1* (8), *wago-1* (11), *ppw-2* (5), *wago-4* (5), and *hrde-1* (4) (*Figure 6E*). We also observed substantially reduced fertility that took longer to manifest in *alg-3* and *alg-4* single mutants (*Figure 6E*). These *agos* have been shown to act partially redundantly (*Conine et al., 2010*) and loss of both results in sterility at 25°C (*Figure 1E*). Collectively, these findings implicate 11/19 AGOs and every type of sRNA pathway in maintaining fertility over generations.

Understanding of the Mrt phenotype is currently incomplete. Factors involved in telomere and chromatin maintenance as well as germ granule components result in an Mrt phenotype when mutated, and it has been suggested that this phenotype occurs when epigenetic stress accumulates over generations (*Cecere, 2021*; *Sundby et al., 2021*). The *hrde-1* Mrt phenotype at 25°C was shown to be reversible upon transferring the animals back to 20°C, suggesting it may not be genotoxic in origin (e.g., transposon mobilization) (*Spracklin et al., 2017*). Similarly, the Mrt phenotype of *prg-1* mutants is reversible when the animals are subjected to starvation (*Simon et al., 2014*). We therefore explored whether *ago* mutant Mrt phenotypes display the same characteristics as each other and

other Mrt mutants. We focused on the WAGOs that have the most severe Mrt phenotype: *wago-1*, *ppw-2*, *wago-4* and *hrde-1*.

To test whether genotoxic stress is responsible for the Mrt phenotype, we maintained animals at 25°C for three generations, then transferred them back to 20°C and measured brood size at each generation. For all four mutants, we observed the same pattern; a reversal of the Mrt phenotype (*Figure 6—figure supplement 2A*), suggesting that genotoxic stress is not responsible for the Mrt phenotype for any of these w*agos*.

Given that the Mrt phenotype results in a gradual decrease in brood size, we hypothesized that the germline might suffer from general proliferation defects. To test this, we examined the germlines of the Mrt *wagos,* focusing on the mitotic zone. Counting germ cells at the mitotic zone at 20°C and 25°C revealed Mrt *wagos* produced fewer germ cells than wild-type controls, suggesting a proliferation defect (*Figure 6—figure supplement 2B*). The Mrt *wagos* also showed a reduced number of oocytes in adult animals (*Figure 6—figure supplement 2C*). We also tested whether the Mrt *wagos* displayed increased levels of apoptosis, which could lead to decreased numbers of germ cells, using acridine orange staining. Only *hrde-1* mutants displayed a slight elevation in apoptosis levels (*Figure 6—figure supplement 2C*). These results are consistent with a previous observation that *hrde-1* mutants have variable defects in both oogenesis and spermatogenesis (*Buckley et al., 2012*) and suggest that the defects observed in the Mrt *wagos* generally arise early in germ cell proliferation. In our analysis of the sRNA complements and targets of these WAGOs, we showed that WAGO-1, PPW-2, and HRDE-1 cluster together to target repetitive elements and other silenced germline genes (*Figure 5D and E*), while WAGO-4 mainly targets germline expressed genes that are co-targeted by CSR-1 (*Figure 5A and C*). Whether a common mechanism or different mechanisms underlie the Mrt phenotype in these *ago* mutants from different clusters remains to be resolved.

## AGO somatic expression and tissue-specific gene regulation

We found broad expression patterns for many AGOs in the soma, suggesting they may have gene regulatory roles in tissues where sRNA pathways have not been deeply explored (*Figure 7F*, *Figure 7—figure supplements 1–8*). The intestine is a key interface between the worm and its environment. Foreign RNA from bacterial food or pathogens, such as viruses, can enter the worm via the intestinal epithelium (*Franz et al., 2014*; *Braukmann et al., 2017*). The intestine is the somatic tissue that expresses the most AGOs: ALG-1, ALG-2, ERGO-1, RDE-1, CSR-1a (the long isoform of CSR-1), PPW-1, SAGO-2, SAGO-1, and NRDE-3 (*Figure 7A and B*, *Figure 7—figure supplement 8*; *Guang et al., 2008*; *Vasquez-Rifo et al., 2013*; *Charlesworth et al., 2021*). Most AGOs were broadly cytoplasmic; NRDE-3 was nuclear; and three AGOs, SAGO-1, SAGO-2, and PPW-1, showed an accumulation along the apical membrane of the intestinal cells (*Figure 7B*). SAGO-2 is not detectable in other tissues (*Figure 7—figure supplements 1–8*).

Given the role of the intestine, and the abundance of AGOs representing three of the four sRNA pathways (miRNA, 22G- and 26G-RNAs), we wondered how these pathways might function there. The entire ERGO-1 cluster is represented in the intestine, along with the long isoform of CSR-1, CSR-1a (*Figure 7A*). Our previous work demonstrated that CSR-1a silences genes involved in immune and pathogen defense responses in the intestine. Loss of *csr-1a* led to an upregulation of these genes and enhanced worm survival on the bacterial intestinal pathogen *Pseudomonas aeruginosa* (PA14) (*Charlesworth et al., 2021*). Similarly, the GO terms associated with the ERGO-1 cluster are enriched for immune and defense responses, particularly in the intestine (*Figure 7C*, *Supplementary file 7*). GO terms associated with the targets of individual AGOs were primarily derived from the targets of SAGO-2 (*Supplementary file 7*), which is closely related to PPW-1 and SAGO-1 (*Figure 1A*). Additionally, PPW-1 targets were enriched for the innate immune signaling pathway, MAP kinase (MAPK) (*Supplementary file 7*; *Kim et al., 2002*). Given their localization and GO analysis, we hypothesized that SAGO-1, SAGO-2, and PPW-1 may play a role in immune responses to pathogenic bacteria. In agreement, survival assays of *sago-2*, *ppw-1,* and *sago-1* mutants on PA14 revealed that both *sago-2* and *ppw-1* are partially resistant to killing by PA14 infection (*Figure 7D*).

Previous research has shown that PPW-1, SAGO-1, and SAGO-2 act as secondary AGOs, downstream of RDE-1 in exo-RNAi (*Yigit et al., 2006*), with SAGO-1 and SAGO-2 being required for efficient exo-RNAi in the soma, and PPW-1 for efficient exo-RNAi in the germline. Overexpression of any of these three AGOs in the muscle cells of a compound *ago* mutant rescues the RNAi deficiency of the

mutant (*Yigit et al., 2006*). RDE-1 and other components of the exo-RNAi machinery are also involved in antiviral responses against the Orsay virus. RDE-1 targets the viral RNA and recruits the RdRP RRF-1 to generate secondary 22G-RNAs that combat the virus. Consistent with this, loss of *rde-1* or the RdRP machinery leads to increased viral proliferation (*Félix et al., 2011*).

Given our understanding of the roles of the secondary AGOs in exo-RNAi, we hypothesized that viral secondary 22G-RNAs could be loaded into these AGOs, and that their mutation could render the worms more sensitive to Orsay virus. To test this, we analyzed the response of *sago-2*, *ppw-1*, and *sago-1* mutants to infection by Orsay virus using *rde-1* and *pals-22* as controls for sensitivity and resistance, respectively (*Reddy et al., 2019*; *Figure 7E*). Only *sago-2* showed a higher infection level, which phenocopied that of *rde-1* mutants (*Figure 7E*). Thus, SAGO-2 may have dual roles in mediating immunity in the intestine. First, SAGO-2 targets immune response genes, and loss of *sago-2* enhances the ability of the worms to survive on PA14. Conversely, loss of *sago-2* decreases viral RNA targeting, likely by RNAi mechanisms in which SAGO-2 is loaded with secondary 22G-RNAs that were generated after targeting of viral RNA by the primary AGO RDE-1.

## Discussion

Here, we have analyzed the 19 AGO proteins in *C. elegans* using CRISPR-Cas9 genome editing and next-generation sequencing. Analysis of the expression patterns and sRNA complements of each AGO identifies sRNA regulatory networks employed in tissues throughout the animal and reveals specific and shared features of each AGO, advancing understanding of the functions and mechanisms of these pathways in the context of a whole animal.

### *C. elegans* small RNA pathways

Our analysis provides a framework for categorizing the AGOs and their sRNAs. Consistent with current models, the AGOs can be divided into four groups based on the type of sRNA they interact with: (1) the miRNA binding classical AGOs, ALG-1, ALG-2, ALG-5, and RDE-1; (2) the piRNA binding PIWI, PRG-1; (3) the 26G-RNA binding classical AGOs, ERGO-1, ALG-3, and ALG-4; and (4) the 22G-RNA binding WAGOs, CSR-1, VSRA-1, WAGO-1, PPW-2, WAGO-4, PPW-1, SAGO-2, SAGO-1, HRDE-1, and NRDE-3. Our analysis of the 22G- and 26G-RNA binding AGOs revealed they can be further classified into four major clusters based on their targets: (1) the CSR-1 cluster: CSR-1, WAGO-4, and C04F12.1/VSRA-1, which target germline expressed protein-coding genes; (2) the WAGO cluster: WAGO-1, PPW-2, and HRDE-1 that target silenced germline genes, pseudogenes, and repetitive/transposable elements; (3) the ERGO-1 cluster: ERGO-1, PPW-1, SAGO-1, SAGO-2, and NRDE-3 that target many somatic genes, pseudogenes, and lincRNAs; and (4) the ALG-3/4 cluster: ALG-3, ALG-4, and WAGO-10 that are restricted to the spermatogenic germline and predominantly target spermatogenesis genes.

Among these groups, several AGOs bind to one type of sRNA (e.g., ALG-1, -2, -5, and PRG-1), while others have broader specificity or act as scavengers (e.g., RDE-1). We observed a physical association between the 26G-RNA binding AGOs and miRNAs, which may reflect coordinated regulation of transcripts by both miRNAs and 26G-RNAs. The 22G-RNA binding AGOs represent a more varied group, in which some AGOs cluster differently depending on the portion of the genome they target (e.g., VSRA-1 clusters with CSR-1 to target protein-coding genes, and with ERGO-1 to target lincRNAs). However, how sRNAs and targets are 'selected' by different AGOs remains poorly understood.

### AGO/small RNA specificity

Individual AGO target specificity involves many factors, including the intrinsic structural and biochemical properties of the AGO; the sRNA biogenesis mechanisms and biochemical features; the features and co-factors of target transcripts; and the expression and localization patterns of AGOs, sRNA machinery, and targets. For instance, it has been shown that the preference for a specific 5′ nucleotide is determined in large part by interactions between the sRNA, the 5′ binding pocket within the MID domain and another region of the AGO termed the specificity loop (*Ma et al., 2005*; *Frank et al., 2010*). In *C. elegans*, the biogenesis mechanisms of sRNAs contribute to specificity through differences in 5′ nucleotide chemistry, in which miRNAs, piRNAs, and 26G-RNAs possess a monophosphate (DICER-dependent), whereas the 22G-RNAs possess a triphosphate (DICER-independent).

This difference is reflected in the AGO phylogeny; the AGOs more closely related to AGOs in other species retained the ability to bind 5′ monophosphorylated nucleotides and possess highly conserved residues within the 5′ binding pocket (e.g., Y529, K533, Q545, K570, C546 in hAGO2) while the WAGOs preferentially bind triphosphorylated nucleotides and possess a more divergent set of residues in these positions. These results, coupled with structure-function-based analyses in vivo, will enable us to understand the mechanisms of AGO loading and sRNA preference more comprehensively despite a lack of in vitro sRNA loading assays.

We observed that most of the transcriptome, including nearly all protein-coding genes, has 22G- and 26G-RNAs directed against it. This suggests that sRNA production (1) happens broadly across tissues and (2) that most of the transcriptome has the potential to become a substrate for RdRP activity. However, we did not observe sRNA enrichment in AGOs for all of the transcripts with detectable sRNAs. This could be for a number of reasons. First, our experimental design and analysis pipeline is biased for more abundant and more enriched sRNAs to produce a high-confidence set of targets for each AGO. If we reduced the thresholds of 5 RPM, twofold enrichment, and requirement for enrichment in both replicates, we might detect additional sRNAs (and thus targets) enriched in association with AGOs that occur at very low abundance or are expressed in a small number of cells. Second, features of the transcript are likely to influence the extent to which it can serve as a template for sRNA synthesis, including sequence motifs, intron/exon, content, 3′ UTR and poly-A length, secondary structure, expression level, association with other RBPs, and subcellular routing and/or localization. Third, targeting by an AGO is generally thought to initiate an sRNA amplification loop. Thus, targets for which sRNAs are successfully loaded into AGOs and that reach a critical threshold of AGO regulation may be the 'winners' that we have detected in our sRNA sequencing. In-depth computational analyses of the features of high-confidence target transcripts will reveal specific features of 'successful' targets. Measuring transcript levels and localization in specific cell types, using single cell-seq and smFISH, will inform our understanding of what target levels and which subcellular locations (e.g., germ granules) are associated with high levels of sRNAs. Finally, examination of RdRP mechanisms in specific cell types and in vitro is necessary to fully understand AGO/sRNA specificity.

## Temporal and spatial specificity of small RNA regulation

Our analysis maps the expression patterns of every AGO throughout development. While some AGOs are broadly expressed in a variety of tissues, others are restricted to specific tissues, cell types, and developmental stages. For example, ERGO-1, ALG-2, and NRDE-3 are expressed in similar patterns within a variety of tissues that include neurons, the somatic gonad, intestine, and oocytes. PRG-1, which has mainly been studied for its role in germline gene regulation, is also present within muscle during larval stages. It remains to be determined whether AGOs have the same functions and targets in each cell type in which they are expressed, and whether the targets change during development.

Our analysis mainly focused on the L4 to YA transition stage in *C. elegans* because all AGOs, aside from the spermatogenesis specific AGOs, are expressed at this time. Also, sRNA populations change in abundance during development (*Ambros et al., 2003*; *Ruby et al., 2006*), which could reflect changes in expression or association with AGOs. Using the tools developed here, it is possible to probe different life stages of *C. elegans* to observe the temporal dynamics of AGO/sRNA complexes and gain a better understanding of the regulation of targets throughout development, either in a whole animal or a tissue/cell-type-specific manner.

Several studies have analyzed cell-type-specific functions of AGO/sRNA pathways. However, most genomic studies on *C. elegans* AGOs and sRNAs used whole worms to obtain sufficient material for IP/sRNA sequencing and mainly considered two tissue types (soma and germline) using mutants and subtractive approaches. While using whole worms enables a broad overview of AGO/sRNA targets, it may miss low-abundance sRNAs that could participate in cell-type-specific functions. We are now able to identify AGO complexes and pools of sRNAs in specific cells or tissues with low amounts of starting material, and can use tissue-specific promoters and 3′ UTRs to drive AGO expression in specific cells and tissues. Furthermore, functional assays, such as reporter assays, are growing increasingly more precise, and coupling these with auxin-degron-mediated AGO depletion (*Zhang et al., 2015*) will allow for enhanced control over AGO activity in specific tissues. Our analysis of expression patterns provides an atlas of AGO expression. This and the phenotypes we have uncovered point to specific tissues of interest, and will help prioritize specific cells and tissues for subsequent analyses.

## Stress reveals phenotypes

*C. elegans* laboratory culture conditions are chosen to minimize stress and promote growth. The natural environment for *C. elegans* presents a much more challenging set of conditions to which the worm must continually adapt. Previous studies did not observe phenotypes for most single *ago* mutants under normal laboratory culture conditions (*Yigit et al., 2006*). However, one major function of sRNA pathways is to regulate gene expression to ensure robustness against stressful conditions, with a growing body of literature demonstrating that sRNA pools are altered in response to changes in the environment (*Rechavi et al., 2011*; *Rechavi et al., 2014*; *Kaletsky et al., 2020*; *Moore et al., 2019*; *Houri-Zeevi et al., 2020*; *Ewe et al., 2020*; *Schott et al., 2014*; *Ni et al., 2016*). For example, in sRNA pathway and germ granule mutants, elevated temperature strongly affects *C. elegans* fertility, leading to an Mrt phenotype (*Ahmed and Hodgkin, 2000*; *Sundby et al., 2021*). While several *agos* had been associated with the Mrt phenotype previously, our systematic analysis revealed additional AGOs whose loss also contributed to reduced fertility under temperature stress and implicated all types of worm sRNA pathways in this process. Therefore, the molecular mechanisms underlying this phenotype may be different for each *ago* mutant. Further in-depth characterization of germline development and gene expression in the *ago* mutants will be necessary to better understand this phenomenon. However, the potential for indirect effects due to mis-regulation of large groups of genes remains a challenge to disentangle.

Pathogens are another set of stressors that worms are frequently exposed to in the wild, but rarely encounter in the lab. Our sequencing analysis of SAGO-2- and PPW-1-bound sRNAs revealed that these AGOs regulate immunity and pathogen response genes, which led us to test their roles in response to various pathogens. Our studies revealed a dichotomy for two closely related paralogs. These two AGOs share greater than 98% sequence identity at the nucleotide level and the same expression pattern in the intestine, yet PPW-1 is also expressed constitutively in the germline. Loss of either *sago-2* or *ppw-1* led to enhanced survival when confronted with the bacterial pathogen PA14, yet loss of *sago-2* alone resulted in increased Orsay virus infection. We suspect this is because SAGO-2 and PPW-1 regulate immune-responsive genes, and their loss is likely to cause mis-regulation of these genes. This may indirectly provide the worms with an enhanced ability to ward off infection by PA14. On the other hand, SAGO-2 is likely to be directly involved in the antiviral response, downstream of RDE-1, while PPW-1 is either redundantly required or dispensable for this response. These two AGOs display different phenotypes, yet vary at only 12 amino acids, and challenging us to understand the molecular mechanisms by which these AGOs act.

While we do not know all the conditions under which AGO/sRNA pathways are required, the worm's natural environment may provide important clues and experimental contexts for further analyzing the roles of the AGOs. This, in combination with the AGO expression profiles and sRNA sequencing data described here will help define environmental stressors and the functions of sRNA pathways in adapting to them.

## Noncoding RNA targets of sRNA pathways

Transposable elements are tightly regulated in the germline. In many animals, transposable elements are silenced by the piRNA pathway. However, in worms the piRNA pathway directly regulates only a handful of DNA (cut and paste) transposable elements. Instead, the WAGO cluster, including HRDE-1, WAGO-1, PPW-1, and PPW-2, is responsible for nearly all transposable element regulation in the worm. These AGOs likely function constitutively in the germline, while a handful of transposable elements are regulated during spermatogenesis by ALG-3/4.

While sRNA regulation of protein-coding genes and transposable elements is well-studied, we show that lincRNAs and pseudogenes are also prominent targets. Both pseudogenes and lincRNAs appear to be regulated by various AGO clusters, implying tissue-specific regulation. For instance, the WAGO cluster targets germline lincRNAs, the ALG-3/4 cluster targets spermatogenesis lincRNAs, and the ERGO-1 cluster targets somatic lincRNAs, when integrating sRNA enrichment and AGO expression data. Both lincRNAs and pseudogenes have the capacity to regulate gene expression themselves (*Pink et al., 2011*; *Statello et al., 2021*), and warrant further study as sRNA targets.

### AGO isoforms and differential functions

Recent studies on two CSR-1 isoforms demonstrated that different isoforms encoded from a single *ago* gene can have different expression patterns, sRNA binding partners, and functions (*Charlesworth et al., 2021*; *Nguyen and Phillips, 2021*). SAGO-2, PPW-1, ALG-1, ALG-2, and ERGO-1 also exhibit the potential to encode more than one protein, with one or more exons that vary between isoforms. For all but SAGO-2, the differential exons are encoded at the 5′ end of the gene, leading to N-terminal differences in encoded proteins. The reagents we generated tag both isoforms of ALG-1, ALG-2, and SAGO-2. However, we have only tagged the longest isoforms of ERGO-1 and PPW-1. Future studies will be needed to identify and characterize any additional functions of distinct AGO isoforms.

### Conclusion

Our work provides a framework for understanding the complete portrait of sRNA biology in the worm, which is a long-standing model for sRNA research. Our studies point to tissue-specific roles for AGOs in regulating particular facets of the genome, highlight networks of AGO function, and reveal novel, stress-linked phenotypes when these pathways are lost. This knowledge provides a basis for elaborating detailed mechanistic insights and opens new avenues of research into AGO and sRNA function.

## Materials and methods

### Materials availability statement

Further information and requests for resources and reagents should be directed to and will be fulfilled by the lead contact, Julie Claycomb (julie.claycomb@utoronto.ca). New strains created in this study have been deposited at the Caenorhabditis Genetics Center. All high-throughput sequencing data are available through GEO, accession number GSE208702. Custom R scripts are available via GitHub: https://github.com/ClaycombLab/Seroussi_2022 (copy archived at *Seroussi, 2023*).

### Experimental models and subject details

#### *C. elegans* strains

A complete list of strains used in this study is provided in *Supplementary file 8*. The Bristol strain N2 was used as the reference strain.

#### Nematode growth

All strains were maintained at 20°C unless otherwise indicated on 3.5 cm plates containing Nematode Growth Medium (NGM) seeded with *Escherichia coli* OP50 bacteria as a food source (*Brenner, 1974*).

### Method details

#### Phylogenetic tree construction

Protein sequences of Argonautes were aligned using MUSCLE with default settings (*Madeira et al., 2019*). Evolutionary analyses were conducted in MEGA X (*Kumar et al., 2018*). The evolutionary history was inferred using the maximum likelihood method. The initial tree for the heuristic search was obtained by applying Neighbor-Join and BioNJ algorithms to a matrix of pairwise distances estimated using a JTT model, and then selecting the topology with superior log likelihood value. A discrete Gamma distribution was used to model evolutionary rate differences among sites (two categories [+G, parameter = 1.9757]). The rate variation model allowed for some sites to be evolutionarily invariable ([+I], 0.62% sites). The bootstrap consensus tree was inferred from 1000 replicates. Branches corresponding to partitions reproduced in less than 50% bootstrap replicates are collapsed.

#### CRISPR/Cas9 genome editing

Tagging genes with GFP::3xFLAG was conducted as previously described (*Dickinson et al., 2015*). Single-guide RNAs (sgRNAs) were designed using CRISPOR (*Concordet and Haeussler, 2018*) and cloned into pDD162 via site-directed mutagenesis PCR (see *Supplementary file 8* for the primers used). For repair templates, homology arms of 500–700 bp on either side of the insertion site were amplified using Q5 High Fidelity Polymerase from N2 genomic DNA and cloned into pDD282 cut with ClaI and SpeI restriction sites using NEBuilder HiFi Assembly mix. The homology arms for *ppw-1* and

*sago-2* were amplified from *sago-2(tm894)* and *ppw-1(tm914)* genomic DNA, respectively, since these harbor deletions allowing for the design of primers that will specifically amplify the intended genomic regions (*ppw-1* and *sago-2* are highly similar in sequence). If the sgRNA site was not destroyed by the insertion of the repair template, synonymous mutations were introduced into the PAM sequence or 3–4 synonymous mutations were introduced into the sgRNA sequence (see *Supplementary file 8* for the primers used). Inserts of all cloned plasmids were verified by Sanger sequencing. An injection mix was used as follows: 10 ng/µl repair template, 50 ng/µl sgRNA, 10 ng/µl pGH8, 5 ng/µl pCFJ104, 2.5 ng/µl pCFJ90.

Tagging genes with 3xFLAG alone was conducted as previously described (*Dokshin et al., 2018*). sgRNAs were designed using CRISPOR (*Concordet and Haeussler, 2018*). *Streptococcus pyogenes* Cas9 protein and guide RNAs (tracrRNA and crRNA) were ordered from IDT. The 3xFLAG repair template was ordered from IDT as an ultramer with ~35 bp of homology arms flanking the insertion site (see *Supplementary file 8*): 5′ 35bp-flanking-region- GATTATAAAGACGATGACGATAAGCGTGACTA CAAGGACGACGACGACAAGCGTGATTACAAGGATGACGATGACAAGAGA-35bp-flanking-region 3′. The pRF4 *rol-6(su1006)* injection marker was used to screen for successfully injected worms. These were screened via PCR flanking the intended insertion site to search for integrations. An injection mix was used as follows: Cas9 250 ng/µl, tracrRNA 100 ng/µl, crRNA 56 ng/µl, 220 ng/µl repair template, and 20 ng/µl pRF4. The 3xFLAG::WAGO-1 strain was generated as described for the GFP::3xFLAG procedure but the GFP sequence was cloned out of the pDD282 plasmid, leaving only the 3xFLAG (see *Supplementary file 8* for primers used).

To generate indels, sgRNAs spanning a genomic region were designed and injected. Mutation of the *dpy-10(cn64)* gene at the same time was used as a co-CRISPR marker to identify and enrich candidate editing events (*Arribere et al., 2014*). An injection mix was used as follows: 20 ng/µl *dpy-10* conversion template, 50 ng/µl sgRNA, 10 ng/µl pGH8, 5 ng/µl pCFJ104, and 2.5 ng/µl pCFJ90. Candidate mutants were screened via PCR spanning the genomic region to be excised. This method was used to generate the *wago-10(tor133)* allele that deletes the region between the 695 nt and the 2394nt (1699 bp deletion).

To generate single-nucleotide polymorphisms, a similar approach to generating 3xFLAG insertions was used where the repair template oligo was ~100 bp of the genomic sequence with mutations to insert with the sgRNA cut site in the middle. With this approach the *sago-2(tor135)* allele was generated where the start methionine and eighth amino acid were changed to stop codons (see *Supplementary file 8*).

## Brood size assays

Five L4 hermaphrodites were transferred to a 15 mm NGM plate seeded with OP50 and allowed to propagate at the desired temperature (20 or 25°C). The progeny of these animals were used in the brood size assay. An individual L4 hermaphrodite was transferred to a 15 mm plate and transferred to a fresh plate every day until egg laying ceased (typically 3 days at 25°C and 4 days at 20°C). The hatched progeny were counted. At least 10 hermaphrodites were assayed per strain.

## Mortal germline assays

The assays were performed similarly to *Ahmed and Hodgkin, 2000*. Five L4 worms were picked to five individual plates to establish five lines and incubated at 25°C. Each generation (every 3 days) five L4 worms were picked from each plate to a new plate. A line was considered mortal if there were no more progeny to pick five L4s from.

## RNAi

RNAi by feeding was conducted as described (*Kamath et al., 2001*; *Ahringer, 2006*). Three L4 worms were placed on NGM plates supplemented with 25 µg/ml carbenicillin and 1 mM IPTG and seeded with the specific RNAi bacterial strain. The bacteria were grown overnight in LB supplemented with 100 µg/ml carbenicillin. The progeny of these worms were tested for the expected RNAi phenotype.

## PA14 survival assays

*P. aeruginosa* (PA14) was streaked on standard LB plates supplemented with carbenicillin at 100 μg/ml and grown overnight. Single colonies were picked and grown in 3 ml of LB overnight culture. Also, 20 μl of PA14 was seeded on 3.5 cm slow killing (SK) NGM plates as previously described (*Mahajan-Miklos et al., 1999*; *Tan et al., 1999*). These SK plates were subsequently incubated overnight at 37°C and then equilibrated for 2 days at 25°C. All strains used for the PA14 survival assay were grown to gravid adults on 3.5 cm NGM plates at 20°C and bleached. The progeny that survived bleaching were then grown to the L4 stage on NGM plates at 20°C. 50 L4s were then plated on SK plates in technical triplicates and subsequently moved to 25°C. Worms were transferred to new SK plates every 24 hr. Counts of the number of dead worm carcasses were performed after 48 hr prior to transferring and performing a final count of both dead worm carcasses and live worms after 72 hr.

## Orsay virus infection assay

Orsay virus filtrate was prepared as previously described (*Bakowski et al., 2014*). Plates of Orsay virus-infected worms were maintained until starvation. Virus from infected worms was collected by washing plates with M9, passing through 0.22 μm filters (MilliporeSigma), and stored at −80°C. Next, ~1000 L1 worms were mixed with 100 μl of 10× OP50-1 and 500 μl of the viral filtrate and then plated on 6 cm NGM plates. At 16 hr post infection (hpi), animals were fixed and fluorescent in situ hybridization (FISH)-stained to assess infection status. Worms were fixed in 1 ml of 4% paraformaldehyde (PFA) in PBS containing 0.1% Tween20 (PBST), for 20 min at room temperature (RT) or overnight at −20°C. Worms were then washed once in 1 ml hybridization buffer (900 mM NaCl, 20 mM Tris [pH 8.0], and 0.01% SDS) and incubated overnight at 46°C in 100 μl hybridization buffer containing FISH probe (5–10 ng/μl) conjugated to a Cal Fluor 610 dye (LGC Biosearch Technologies). Orsay Probe 1 (gaca tatgtgatgccgagac) and Orsay Probe 2 (gtagtgtcattgtaggcagc) were mixed at a 50:50 (10 ng/μl) ratio and used to detect Orsay virus. Stained animals were washed once in 1 ml wash buffer (hybridization buffer containing 5 mM EDTA) and incubated in 500 μl fresh wash buffer for a further 30 min at 46°C. Worms were resuspended in 20 μl EverBrite Mounting Medium (Biotium) and mounted on slides for imaging. Worms with any number of cells stained with the FISH probes were considered infected.

## Protein lysate preparation

Synchronous populations of ~100,000 L1 worms were plated on 15 cm NGM plates with ~2 ml of 5× concentrated OP50 *E. coli* as a food source. Five of these plates were used as starting material for protein isolation. Worms were grown for 48 hr for L4 staged worms or 58 hr for young adults (worms that had transitioned to producing oocytes but not yet with embryos). Worms were washed three times with M9 buffer (22 mM KH$_2$PO$_4$, 42 mM Na$_2$HPO$_4$, 86 mM NaCl) and one time with EDTA buffer (10% glycerol, 10 mM EDTA, 30 mM HEPES, 100 mM potassium acetate). The pellet was flash-frozen in a dry ice/ethanol bath. The frozen pellets were stored at −80°C.

The frozen pellet was resuspended 1.5:1 (v/v) in ice-cold IP buffer (10% glycerol, 10 mM EDTA, 30 mM HEPES, 100 mM potassium acetate, 2 mM DTT, 0.1% NP-40) supplemented with protease and phosphatase inhibitors (one tablet per 5 ml buffer of cOmplete, mini, EDTA-free protease inhibitor cocktail [Roche], 1:100 phosphatase inhibitor cocktail 2 [Sigma], 1:100 phosphatase inhibitor cocktail 3 [Sigma]). If the pellet was to be used for RNA purification, 1% (v/v) SuperaseIN RNase inhibitor (Thermo Fisher) was added. The pellet was homogenized using a stainless steel dounce homogenizer (Wheaton Inc) until intact worms were no longer visible. Extracts were centrifuged at 13,000 × *g* for 10 min at 4°C, and the supernatant transferred to a fresh tube. A Lowry assay was performed to determine total protein concentration using a NanoDrop 1800C spectrophotometer (Thermo Fisher).

## Immunoprecipitation of Argonaute complexes

All IPs were conducted on 5 mg of total protein per reaction. Input control samples were made by taking 10% of the lysate before the addition of antibodies. For the IP of GFP tagged Argonautes, GFP-Trap_MA beads (ChromoTek) were equilibrated by washing them three times in 1 ml of IP buffer. Then, 20 μl of beads were added to 5 mg total protein in a reaction volume of 500 μl and rotated at 4°C for an hour. The beads were then washed four times with 1 ml of IP buffer, separated from the supernatant on a magnetic stand, and rotated for 10 min at 4°C between each wash. For the IP of

3xFLAG tagged Argonautes, Monoclonal Anti-FLAG M2 (Sigma-Aldrich) were bound to Dynabeads Protein G (Thermo Fisher) or GB-Magic Protein A/G Immunoprecipitation Magnetic Beads according to the manufacturer's instructions. Then, 5 µg of Anti-FLAG M2 in 200 µl of PBS-T were added to 50 µl of Dynabeads and rotated at RT for 10 min. Also, IPs were conducted as described above except the whole 50 µl of ANTI-FLAG M2-bound beads were used in the IP. For IPs treated with RNase (e.g., ERGO-1), 100U of RNase I (Thermo Fisher) was used at 37°C for 15 min.

For small RNA sequencing, 3–6 IP reactions were combined in 200 µl of IP buffer. Then, 800 µl of TRI Reagent (Molecular Research Centre) was added, and the samples were frozen at –80°C until RNA extraction and sequencing were done as described below.

## Western blot analysis

Proteins were resolved by SDS-PAGE on precast gradient gels (4–12% Bis-Tris Bolt gels, Thermo Fisher) and transferred to Hybond-C membrane (Amersham Biosciences) using a Bio-Rad semi-dry transfer apparatus at 25 V for 45 min. The membrane was washed three times with PBST (137 mM NaCl, 2.7 mM KCl, 10 mM $Na_2HPO_4$, 1.8 mM $KH_2PO_4$ pH 7.4, 0.1% Tween-20) and blocked with 5% milk-PBST (PBST, 5% skim milk) for 1 hr at RT. The membrane was then incubated overnight at 4°C with 1:2000 M2 Anti-FLAG antibody (Sigma-Aldrich) in 5% milk-PBST. The membrane was washed three times with PBST and then blocked with 5% milk-PBST for 30 min at RT. The membrane was incubated with 1:1000 anti-mouse IgG HRP-linked antibody (Cell Signaling Technology) in 5% milk-PBST for 1 hr at RT. The membrane was washed three times in PBST and then developed using Luminata Forte Western HRP substrate (Millipore).

## RNA isolation

RNA was isolated using TRI Reagent (Molecular Research Centre). Samples were mixed with TRI Reagent in a 1:4 ratio and frozen at –80°C. Samples were vortexed at RT for 15 min and frozen again at –80°C for 15 min. This was repeated for a total of three times. Then, 100 µl of chloroform was added to the samples and centrifuged at 12,000 × $g$ for 15 min at 4°C. The top aqueous phase was transferred to a fresh tube. Phenol:chloroform:isoamyl alcohol (Sigma-Aldrich) was added in a 1:1 ratio and centrifuged at 12,000 × $g$ for 15 min at 4°C. The top aqueous phase was transferred to a fresh tube. Then, 20 µg of glycogen (Ambion) and a 1:1 ratio of isopropanol were added to the samples and incubated at –20°C for 30 min. Samples were then centrifuged at 16,000 × $g$ for 30 min at 4°C and the supernatant was removed. The pellet was washed with 900 µl of 70% ice-cold ethanol for 10 min and centrifuged at 16,000 × $g$ for 10 min at 4°C. This was repeated twice. The pellet was then left to air dry for 10 min at RT and then resuspended in 6–25 µl of RNase-free water preheated to 70°C.

## Small RNA library preparation and sequencing

Small RNA libraries were prepared with the NEBNext Small RNA Library Prep Set for Illumina (New England Biolabs) following the protocol provided by the manufacturer. Then, 1 µg of total RNA or immunoprecipitated RNA was used as starting material. The resulting DNA library was visualized using 8% PAGE and bands corresponding to small RNAs of size range 16–30 bp (~135–150 bp on gel) were excised. The DNA was eluted from the excised bands by rotating overnight in 500 µl of DNA Gel Elution buffer at RT. The DNA was precipitated with 20 µg glycogen (Ambion), 50 µl of 3 M sodium acetate pH 5.2, and 1 volume of isopropanol (Sigma-Aldrich) as described above and ultimately resuspended in 12 µl of Ultra-Pure water. The library DNA was quantified using a Qubit HS DNA kit (Thermo Fisher). Between 12 and 19 libraries were pooled in equal amounts and sequenced on a HiSeq 2500 Sequencing System (Illumina).

## Small RNA sequencing analysis

The small RNA sequences obtained from the sequencer were first assessed for quality using FastQC (version 0.11.5, *Andrews, 2010*). Adapter sequences were then removed using cutadapt (version 1.15, *Martin, 2011*) using the following command: -a ADAPTER -f fastq -m 16M 30 `--discard-untrimmed`.

The sequences were then run through FastQC again to assess quality. The trimmed reads were then aligned to the *C. elegans* PRJNA13758 ce11 genome assembly (WormBase version WS276) with STAR (version 2.6.0c, *Dobin et al., 2013*) using the following commands: `--runThreadN` 12

--outSAMtype BAM SortedByCoordinate --outFilterMultimapNmax 50 --outFilterMultimap-ScoreRange 0 --outFilterMismatchNoverLmax 0.05 --outFilterMatchNmin 16 --outFilterScoreMinOverLread 0 --outFilterMatchNminOverLread 0 --alignIntronMax 1.

The reads were then counted using a custom R (version 3.6.3) script against publicly available genome annotations. The WormBase version WS276 PRJNA13758 ce11 canonical geneset annotations were used (excluding miRNAs, repeats, and transposons). *C. elegans* miRNA annotations were obtained from miRBase (release 22.1). For repeats and transposons RepeatMasker+Dfam (ce10, October 2010, RepeatMasker open-4.0.6, Dfam 2.0) annotations were used. The UCSC Lift Genome Annotations tool was used to convert ce10 coordinates to ce11 coordinates. Briefly, the counting script used the findOverlaps function from the GenomicAlignments package (version 1.22.1) to assign reads to features. Multiple aligning reads or reads that align to more than one feature were dealt with by counting reads in a sequential manner to the different gene biotypes in the following order (AS stands for antisense): miRNA, piRNA, rRNA, snoRNA, snRNA, tRNA, ncRNA, lincRNA, repeats AS, protein coding AS, pseudogene AS, lincRNA AS, antisense RNA, rRNA AS, snoRNA AS, snRNA AS, tRNA AS, ncRNA AS, miRNA AS, piRNA AS, protein coding, pseudogene, antisense RNA AS, repeats. For reads that align to more than one feature in the same biotype group, the read count was split between the features based on the fraction of uniquely aligned reads to each of those features (unique weighing). Subsequent analysis was performed using custom R scripts (https://github.com/ClaycombLab/Seroussi_2022, copy archived at *Seroussi, 2023*).

To determine small RNAs enriched in Argonaute IPs, reads were first normalized to library size (reads per million [RPM]). We used two approaches for this. In the first approach, we normalized the reads to the entire library size. In the second approach, we normalized the reads to library size minus sense reads of: rRNA, snoRNA, snRNA, tRNA, ncRNA, lincRNA, protein coding, and pseudogene. These likely represent RNA degradation products so removing them may eliminate noise. The first approach was used in the initial analysis of small RNAs associated with Argonaute IPs for complete transparency and unbiased assignment of small RNA types and targets. Indeed, the vast majority of reads in all libraries were antisense rather than sense. Thus, subsequent enrichment analysis and comparisons between Argonaute IPs and identification of likely targets used the second approach. To determine whether small RNAs against a particular target were enriched in an Argonaute IP, the following calculation was made: enrichment = IP RPM + 0.01/Input RPM + 0.01. The 0.01 represents a pseudocount to eliminate the possibility of dividing by zero. A target was considered enriched if in every replicate it was at least twofold enriched and had at least 5 RPM in the IP replicates. To further refine the analysis, where indicated in the text, only 22G or 26G-RNAs were used to calculate enrichment. These were defined as reads of 20–24 nt and 25–27 nt, respectively, with no 5' nucleotide constraints.

To determine differential expression of small RNAs in mutant *argonaute* strains, we used the R package DESeq2 (version 3.14; *Love et al., 2014*).

Published datasets were used as follows: WAGO-1 small RNA targets and glp-4-enriched/depleted small RNA targets (*Gu et al., 2009*), ERGO-1-enriched small RNA targets (*Vasale et al., 2010*), alg-3; alg-4-depleted small RNA targets (*Conine et al., 2010*), CSR-1-enriched small RNA targets (*Claycomb et al., 2009*), mut-16-depleted small RNA targets (*Phillips et al., 2012*) defined as twofold depleted in mutant and having at least 10 RPM, gamete-specific expressed genes (*Ortiz et al., 2014*), RdRP mutants-depleted small RNA targets were reanalyzed as described above and defined as twofold depleted in mutant and having at least 5 RPM (*Sapetschnig et al., 2015*). All published gene lists used were converted to WS276 gene names using WormBase Converter (*Engelmann et al., 2011*) before being compared to gene lists generated in this study.

GO analysis was performed using gProfiler and Wormbase Enrichment Suite (*Angeles-Albores et al., 2018*; *Supplementary file 7*).

## Microscopy

All images were taken on a Leica DMi8 TCS SP8 confocal microscope, except for those in *Figure 7B*, which were taken on a Nikon TiE microscope with a C2 confocal module. All images presented are a single 0.4 μm slice, taken using a 488 nm laser, and in most instances Normarski/Differential Interference Contrast (DIC)images are also displayed. Images were processed using FIJI, Adobe Photoshop, and Adobe Illustrator.

## Quantification of Argonaute and germ granule factor overlap

Staged GFP::3xFLAG::AGO-expressing worms were washed in M9 and immobilized on positively charged glass slides with 10 µl of 10 mM levamisole. Germlines were dissected with a 17-gauge needle by cutting at the vulva or head/tail. A coverslip was added, and the slides were placed on a flat aluminum block in dry ice for at least 10 min. Slides were either kept at –80°C or fixed immediately. The coverslip was popped off with a razor blade and samples were fixed at –20°C for 5 min in each of, 100% methanol, 50/50 methanol/acetone, and 100% acetone. Samples were air-dried, and a hydrophobic marker was used to outline the sample. All washes and incubations were performed in a humidity chamber (i.e., a lidded plastic tray covered in aluminum foil with wet paper towels and a plastic rack to hold the slides). Samples were washed 2 × 5 min with PBST, then blocked for 30 min at RT with PBST + BSA (1× PBS, 0.1% Tween-20, and 3% BSA). Samples were then incubated with primary antibodies (Anti-HA [Sigma] or anti-PGL) overnight at 4°C. Slides were washed 3 × 10 min with PBST, then blocked with PBST + BSA for 30 min at RT. Samples were incubated with secondary antibodies (anti-rat::TRITC or anti-mouse IgM::TRITC) for 1 hr at RT then washed 3 × 10 min with PBST and 3 × 5 min with PBS. Samples were stained with DAPI (1 µg/ml) for 10 min then washed 3 × 5 min with PBS and mounted in 2 µl of Vectashield (Vector Labs). Samples were kept at –20°C until imaged. Colocalization of proteins was calculated with the ImageJ plugin JaCoP (*Bolte and Cordelières, 2006*). One germline from each of six different animals was imaged per strain and developmental time point. Regions of interest (ROI) were generated using the 3D objects counter plugin in ImageJ (*Schneider et al., 2012*) by adjusting the threshold until only germ granule pixels are detected. Five Z-stacks (0.9 µM apart) were quantified per germline to capture the overlap over a 3D space. Mander's co-localization coefficients are calculated using JaCoP (*Bolte and Cordelières, 2006*).

## Acknowledgements

We are grateful to the Toronto *C. elegans* community for various reagents and helpful discussions, and the lab of Dr. Thomas Hurd for the use of their confocal microscope. We thank members of the Claycomb and Reinke labs and Dr. Guy Riddihough from Life Sciences Editors for feedback on the manuscript.

This work was funded by the Canadian Institutes of Health Research Project Grants PJT-156083, PJG-175378, PJT-178076 to JMC, and PJT-400784 to AWR and Natural Sciences and Engineering Research Council of Canada Discovery Grant RGPIN-2020-06235 to JMC. JMC is a Canada Research Chair Tier II in Small RNA Biology. AWR is supported by an Alfred P Sloan Research Fellowship FG2019-12040. Some strains were provided by the CGC, which is funded by the NIH Office of Research Infrastructure Programs (P40 OD010440).

## Additional information

### Funding

| Funder | Grant reference number | Author |
|---|---|---|
| Canadian Institutes of Health Research | PJT-15608 | Julie M Claycomb |
| Canadian Institutes of Health Research | PJG-175378 | Julie M Claycomb |
| Canadian Institutes of Health Research | PJT-178076 | Julie M Claycomb |
| Canadian Institutes of Health Research | PJT-400784 | Aaron W Reinke |
| Natural Sciences and Engineering Research Council of Canada | RGPIN-2020-06235 | Julie M Claycomb |
| Alfred P. Sloan Foundation | FG2019-12040 | Aaron W Reinke |

| Funder | Grant reference number | Author |
|--------|------------------------|--------|

The funders had no role in study design, data collection and interpretation, or the decision to submit the work for publication.

## Author contributions

Uri Seroussi, Data curation, Software, Formal analysis, Validation, Investigation, Visualization, Methodology, Writing – original draft, Writing – review and editing; Andrew Lugowski, Lina Wadi, Software, Formal analysis, Validation, Methodology; Robert X Lao, Winnie Zhao, Adam E Sundby, Formal analysis, Validation, Investigation, Methodology; Alexandra R Willis, Amanda G Charlesworth, Formal analysis, Validation, Investigation, Methodology, Writing – review and editing; Aaron W Reinke, Conceptualization, Resources, Supervision, Funding acquisition, Visualization, Project administration, Writing – review and editing; Julie M Claycomb, Conceptualization, Resources, Data curation, Supervision, Funding acquisition, Investigation, Visualization, Methodology, Writing – original draft, Project administration, Writing – review and editing

## Author ORCIDs

Aaron W Reinke ![ORCID] http://orcid.org/0000-0001-7612-5342
Julie M Claycomb ![ORCID] http://orcid.org/0000-0003-3132-5206

## Decision letter and Author response

Decision letter https://doi.org/10.7554/eLife.83853.sa1
Author response https://doi.org/10.7554/eLife.83853.sa2

# Additional files

## Supplementary files

• Supplementary file 1. A table summarizing previously published data on *C. elegans* AGOs.

• Supplementary file 2. A table summarizing small RNA library information.

• Supplementary file 3. Tables summarizing the enrichment of reads, biotypes, and AGO targets in all libraries.

• Supplementary file 4. A table summarizing sRNAs depleted in *ago* mutants.

• Supplementary file 5. A table summarizing the mirDeep2 analysis results in predicting novel high confidence miRNAs.

• Supplementary file 6. A table summarizing the 466 unannotated 21U sRNA sequences enriched in PRG-1 IPs and depleted in *prg-1* mutants.

• Supplementary file 7. Gene Ontology analysis results of AGO targets.

• Supplementary file 8. Strains and primers used in this study.

• MDAR checklist

## Data availability

All high-throughput sequencing data are available through GEO, accession number GSE208702. Custom R scripts are available via GitHub: https://github.com/ClaycombLab/Seroussi_2022 (copy archived at *Seroussi, 2023*) *C. elegans* strains generated in this study are available through the Caenorhabditis Genetics Center.

The following dataset was generated:

| Author(s) | Year | Dataset title | Dataset URL | Database and Identifier |
|-----------|------|---------------|-------------|-------------------------|
| Seroussi U, Lugowski A, Wadi L, Lao RX, Willis AR, Zhao W, Sundby AE, Charlesworth AG, Reinke AW, Claycomb JM | 2022 | Data from: A Comprehensive Survey of C. elegans Argonaute Proteins Reveals Organism-wide Gene Regulatory Networks and Functions | https://www.ncbi.nlm.nih.gov/geo/query/acc.cgi?acc=GSE208702 | NCBI Gene Expression Omnibus, GSE208702 |

The following previously published datasets were used:

| Author(s) | Year | Dataset title | Dataset URL | Database and Identifier |
|---|---|---|---|---|
| Conine CC, Batista PJ, Gu W, Claycomb JM, Chaves DA, Shirayama M, Mello CC | 2010 | Data from: Argonautes ALG-3 and ALG-4 are required for spermatogenesis-specific 26G-RNAs and thermotolerant sperm in Caenorhabditis elegans | https://www.ncbi.nlm.nih.gov/geo/query/acc.cgi?acc=GSE18731 | NCBI Gene Expression Omnibus, GSE18731 |
| Vasale JJ, Gu W, Thivierge C, Batista PJ, Claycomb JM, Youngman EM, Duchaine TF, Mello CC, Conte D | 2010 | Date from: Sequential rounds of RNA-dependent RNA transcription drive endogenous small-RNA biogenesis in the ERGO-1/Argonaute pathway | https://www.ncbi.nlm.nih.gov/geo/query/acc.cgi?acc=GSE18714 | NCBI Gene Expression Omnibus, GSE18714 |
| Claycomb JM, Batista PJ, Pang KM, Gu W, Vasale JJ, van Wolfswinkel JC, Chaves DA, Shirayama M, Mitani S, Ketting RF, Mello CC, Conte D | 2009 | Data from: The Argonaute CSR-1 and its 22G-RNA cofactors are required for holocentric chromosome segregation | https://www.ncbi.nlm.nih.gov/geo/query/acc.cgi?acc=GSE18165 | NCBI Gene Expression Omnibus, GSE18165 |
| Gu W, Shirayama M, Vasale JJ, Batista PJ, Claycomb JM, Moresco JJ, Youngman EM, Keys J, Stoltz MJ, Chen CG, Chaves DA, Kasschau KD, Fahlgren N | 2009 | Data from: Distinct argonaute-mediated 22G-RNA pathways direct genome surveillance in the C. elegans germline | https://www.ncbi.nlm.nih.gov/geo/query/acc.cgi?acc=GSE18215 | NCBI Gene Expression Omnibus, GSE18215 |
| Sapetschnig A, Sarkies P, Lehrbach NJ, Miska EA | 2015 | Data from: Tertiary siRNAs mediate paramutation in C. elegans | https://www.ncbi.nlm.nih.gov/geo/query/acc.cgi?acc=GSE66344 | NCBI Gene Expression Omnibus, GSE66344 |

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

# Appendix 1

**Appendix 1—key resources table**

| Reagent type (species) or resource | Designation | Source or reference | Identifiers | Additional information |
|---|---|---|---|---|
| gene (*C. elegans*) | *rde-1* | WormBase | WBGene00004323 | |
| gene (*C. elegans*) | *alg-1* | WormBase | WBGene00000105 | |
| gene (*C. elegans*) | *alg-2* | WormBase | WBGene00000106 | |
| gene (*C. elegans*) | *alg-3* | WormBase | WBGene00011910 | |
| gene (*C. elegans*) | *alg-4* | WormBase | WBGene00006449 | |
| gene (*C. elegans*) | *alg-5* | WormBase | WBGene00011945 | |
| gene (*C. elegans*) | *wago-1* | WormBase | WBGene00011061 | |
| gene (*C. elegans*) | *wago-3/ppw-1* | WormBase | WBGene00004094 | |
| gene (*C. elegans*) | *wago-4* | WormBase | WBGene00010263 | |
| gene (*C. elegans*) | *wago-5* | WormBase | WBGene00022877 | |
| gene (*C. elegans*) | *wago-6/sago-2* | WormBase | WBGene00018921 | |
| gene (*C. elegans*) | *wago-7/ppw-1* | WormBase | WBGene00004093 | |
| gene (*C. elegans*) | *wago-8/sago-1* | WormBase | WBGene00019666 | |
| gene (*C. elegans*) | *wago-9/hrde-1* | WormBase | WBGene00007624 | |
| gene (*C. elegans*) | *wago-10* | WormBase | WBGene00020707 | |
| gene (*C. elegans*) | *wago-11* | WormBase | WBGene00021711 | |
| gene (*C. elegans*) | *wago-12/nrde-3* | WormBase | WBGene00019862 | |
| gene (*C. elegans*) | *csr-1* | WormBase | WBGene00017641 | |
| gene (*C. elegans*) | *C04F12.1/vsra-1* | WormBase | WBGene00007297 | |
| strain, strain background (*C. elegans*; hermaprhodites and males) | Bristol N2 | WormBase | | |
| strain, strain background (*E. coli*) | OP50 | *Caenorhabditis* Genetics Center | | |

*Appendix 1 Continued on next page*

*Appendix 1 Continued*

| Reagent type (species) or resource | Designation | Source or reference | Identifiers | Additional information |
|---|---|---|---|---|
| strain, strain background (*E. coli*) | *HT115* | *Caenorhabditis* Genetics Centers | | |
| genetic reagent (*C. elegans*) | List of strains | This study, **Supplementary file 8**; *Caenorhabditis* Genetics Center | | |
| antibody | Anti-mouse IgG, HRP-linked Antibody (horse polyclonal) | Cell Signaling Technology | 7076 S | 1:1000 for western blots |
| antibody | Monoclonal ANTI-FLAG M2 antibody (mouse monoclonal) | Sigma | F1804 | 1:2000 for western blots 5 µg per 50 µl of Dynabeads in 200 µl for IPs |
| antibody | GFP-Trap_MA (alpaca recombinant nanobody) | ChromoTek | gtma | 20 µl of beads per 5 mg total protein in 500 µl |
| antibody | RFP-Trap_MA (alpaca recombinant nanobody) | ChromoTek | rtma | 20 µl of beads per 5 mg total protein in 500 µl |
| sequence-based reagent | List of oligonucleotides | This study, **Supplementary file 8** | | |
| sequence-based reagent | High Throughput Sequencing Data (**Conine et al., 2010**) | https://www.ncbi.nlm.nih.gov/geo/ | GSE18731 | |
| sequence-based reagent | High Throughput Sequencing Data (**Vasale et al., 2010**) | https://www.ncbi.nlm.nih.gov/geo/ | GSE18714 | |
| sequence-based reagent | High Throughput Sequencing Data (**Claycomb et al., 2009**) | https://www.ncbi.nlm.nih.gov/geo/ | GSE18165 | |
| sequence-based reagent | High Throughput Sequencing Data (**Gu et al., 2009**) | https://www.ncbi.nlm.nih.gov/geo/ | GSE18215 | |
| sequence-based reagent | High Throughput Sequencing Data (**Sapetschnig et al., 2015**) | https://www.ncbi.nlm.nih.gov/geo/ | GSE66344 | |
| sequence-based reagent | High Throughput Sequencing Data | **Phillips et al., 2012** | Table S3 | |
| sequence-based reagent | High Throughput Sequencing Data | **Ortiz et al., 2014** | Table S1 | |
| sequence-based reagent | High Throughput Sequencing Data | **Tzur et al., 2018** | Table S3 | |
| sequence-based reagent | High Throughput Sequencing Data | This study; https://www.ncbi.nlm.nih.gov/geo/ | GSE208702 | |
| commercial assay or kit | NEBuilder HiFi DNA Assembly Cloning Kit | New England Biolabs | E5520 | |
| commercial assay or kit | NEBNext Multiplex Small RNA Library Prep Kit for Illumina | New England Biolabs | E7560 | |

*Appendix 1 Continued on next page*

*Appendix 1 Continued*

| Reagent type (species) or resource | Designation | Source or reference | Identifiers | Additional information |
|---|---|---|---|---|
| chemical compound, drug | Q5 High-Fidelity DNA Polymerase | New England Biolabs | M0491 | |
| chemical compound, drug | T4 DNA Ligase | New England Biolabs | M0202 | |
| chemical compound, drug | Tri Reagent | Molecular Research Centre | TR118 | |
| chemical compound, drug | Phenol Chloroform Isoamyl Alcohol | Sigma-Aldrich | P2069 | |
| chemical compound, drug | cOmplete, Mini, EDTA-free Protease Inhibitor Cocktail | Roche | 11836170001 | |
| chemical compound, drug | RNA 5' Polyphosphatase | Epicentre | RP8092H | |
| chemical compound, drug | Phosphatase Inhibitor Cocktail 2 | Sigma-Aldrich | P5726 | |
| chemical compound, drug | Phosphatase Inhibitor Cocktail 3 | Sigma-Aldrich | P0044 | |
| chemical compound, drug | DTT | BioShop Canada | DTT001 | |
| chemical compound, drug | NP-40 | BioBasic | NDB0385-500ML | |
| chemical compound, drug | Levamisole hydrochloride | Fisher Scientific | AC187870100 | |
| chemical compound, drug | Bovine Serum Albumin | BioBasic | 9048-46-8 | |
| chemical compound, drug | SUPERase• In RNase Inhibitor | ThermoFisher Scientific/ Invitrogen | AM2696 | |
| chemical compound, drug | T4 Polynucleotide Kinase | New England Biolabs | M0201 | |
| chemical compound, drug | Glycogen | Ambion | AM9510 | |
| chemical compound, drug | RNAse and DNAse Away | BioBasic | DB0339 | |
| chemical compound, drug | Dynabeads Protein G | ThermoFisher Scientific/ Invitrogen | 10003D | |
| chemical compound, drug | GB-Magic Protein A/G Immunprecipitation Magnetic Beads | GeneBiosystems | 22202B | |
| chemical compound, drug | Protein Assay Reagent A | Bio-Rad | 5000113 | |
| chemical compound, drug | Protein Assay Reagent B | Bio-Rad | 5000114 | |
| chemical compound, drug | Protein Assay Reagent S | Bio-Rad | 5000115 | |
| chemical compound, drug | Nitrocellulose blotting membrane | GE Healthcare | 10600016 | |
| chemical compound, drug | Luminata Classico Western HRP substrate | Millipore | WBLUC0500 | |

*Appendix 1 Continued on next page*

*Appendix 1 Continued*

| Reagent type (species) or resource | Designation | Source or reference | Identifiers | Additional information |
|---|---|---|---|---|
| chemical compound, drug | RNase I | ThermoFisher Scientific/ Ambion | AM2295 | |
| software, algorithm | MEGA X | https://www.megasoftware.net/ | | |
| software, algorithm | CRISPOR | http://crispor.tefor.net/ | | |
| software, algorithm | SnapGene | https://www.snapgene.com/ | | |
| software, algorithm | GraphPad Prism | https://www.graphpad.com/scientific-software/prism/ | | |
| software, algorithm | STAR | *Dobin et al., 2013* | | |
| software, algorithm | Rstudio | https://www.rstudio.com/ | | |
| software, algorithm | Custom Computational Pipeline | https://github.com/ | https://github.com/ClaycombLab/Seroussi_2022 | |
| other | Bolt Precast Bis-Tris Plus Gradient Gels (4–12%) | ThermoFisher Scientific | NW04120BOX | This study: Materials and Methods |
| other | Hybond C Membrane | GE/Amersham Biosciences | CA95038-380L | This study: Materials and Methods |
| other | Wheaton Steel Dounce Homogenizer | VWR | 62400–675 | This study: Materials and Methods |

