## [Editor Report]

This impressive study presents the most comprehensive analysis of the Argonautes, their small RNA partners, their targets, and their biological functions in any species to date. The work provides new insights into Argonaute-based pathways, includes extensive validation of existing models, and describes overall a treasure-trove of reagents and datasets for future exploration of the vast Argonaute world in *C. elegans*.

---

## [Decision Letter]

**Decision letter after peer review:**

Thank you for submitting your article "A Comprehensive Survey of *C. elegans* Argonaute Proteins Reveals Organism-wide Gene Regulatory Networks and Functions" for consideration by *eLife*. Your article has been reviewed by 2 peer reviewers, and the evaluation has been overseen by a Reviewing Editor and David James as the Senior Editor. The following individuals involved in the review of your submission have agreed to reveal their identity: Amy Pasquenelli (Reviewer #1); Scott Kennedy (Reviewer #2).

*Reviewer #1 (Recommendations for the authors):*

The paper, figures, and tables are clear, engaging and expertly presented as is. I have no suggestions for improvement and applaud the authors for making a huge, intricate study highly accessible and enjoyable to read.

*Reviewer #2 (Recommendations for the authors):*

A few suggestions for improving the manuscript are outlined below. These changes can be made without further experimentation.

The authors show that the identification of ERGO-1 and ALG-1/2 bound small RNAs was confounded by the fact that these AGOs interact with a common set of mRNA targets, resulting in mRNA-dependent cross-contamination in datasets. It is possible that related issues confound other datasets. I recommend the authors acknowledge this issue somewhere in the main text.

I am a bit surprised that N terminal tagging of WAGO-3 resulted in a functional protein. Gudipati et al, 2021 show that WAGO-3 requires sequence-dependent N-terminal processing for functionality. Therefore, I would have thought that N-terminal tagging of WAGO-3 would have disrupted this processing and inactivated WAGO-3. I recommend the authors discuss this issue somewhere in their paper.

---

## [Author Response]

The reviewers have discussed their reviews with one another, and the Reviewing Editor has drafted this to help you prepare a revised submission.Reviewer #1 (Recommendations for the authors):The paper, figures, and tables are clear, engaging and expertly presented as is. I have no suggestions for improvement and applaud the authors for making a huge, intricate study highly accessible and enjoyable to read.

We thank the reviewer for their careful assessment of our work and are grateful that they find the manuscript enjoyable, accessible, and useful for the field.

Reviewer #2 (Recommendations for the authors):A few suggestions for improving the manuscript are outlined below. These changes can be made without further experimentation.

Thank you for your thorough and positive assessment of our work. We find these suggested changes to be an important addition to the manuscript, and have worked to incorporate these points in the text.

The authors show that the identification of ERGO-1 and ALG-1/2 bound small RNAs was confounded by the fact that these AGOs interact with a common set of mRNA targets, resulting in mRNA-dependent cross-contamination in datasets. It is possible that related issues confound other datasets. I recommend the authors acknowledge this issue somewhere in the main text.

We are sorry this did not come through more clearly in the text; it’s a very important point. We have made this point explicitly within this section, on p. 14:

miRNA and ERGO-1 26G-RNA pathways intersect

Collectively, these data suggest that the miRNA enrichment present in the ERGO-1 IPs may be indirect, due to an interaction between ERGO-1 and ALG-1 or ALG-2 on target transcripts. They also imply that co-regulation of target transcripts by 26G-RNAs and miRNAs could occur. Finally, these observations highlight an important consideration for interpreting IP/sRNA sequencing data: the association of multiple AGOs on common transcripts could result in skewed sRNA enrichment patterns, thus additional experiments such as those described above and examination of *ago* mutant sRNA sequencing data are warranted.

While we have tried to minimize non-specific association of AGOs with sRNAs by testing several buffer/IP conditions and performing additional controls (e.g. as for ERGO-1), the issues of AGO-AGO interactions on common transcripts or within complexes, and the possibility of AGOs associating with abundant sRNA species that they do not normally bind during the IP are important considerations for these experiments. We can readily observe this “cross-contamination” issue with ERGO-1 because the sRNAs are two different types: miRNAs and 26G-RNAs. It might not be so easy to pick up on these issues with the WAGOs that bind to 22G-RNAs; for example WAGO-1, HRDE-1, and PPW-2/WAGO-3. However, there are several lines of data that make me less concerned about this issue: 1. The targets are not entirely overlapping for these 3 WAGOs (HRDE-1, WAGO-1, and PPW-2/WAGO-3); and 2. The metagene profiles are different for PPW-2 and HRDE-1 vs. WAGO-1, suggesting that they are not binding to the same exact sequences of sRNAs along the transcript (which you might expect to see if the enrichment was from WAGO-1 and PPW- 2 targeting the same transcripts at the same time, for instance). So, we don’t think this is a huge concern for the WAGOs and we’re not sure why it happens for ERGO-1/ALG-½.

I am a bit surprised that N terminal tagging of WAGO-3 resulted in a functional protein. Gudipati et al, 2021 show that WAGO-3 requires sequence-dependent N-terminal processing for functionality. Therefore, I would have thought that N-terminal tagging of WAGO-3 would have disrupted this processing and inactivated WAGO-3. I recommend the authors discuss this issue somewhere in their paper.

We were also surprised that PPW-2/WAGO-3 was functional, given the tag placement and the findings of Gudipati regarding N-terminal processing by DPF-3. The tag is not inserted at the extreme N-terminus of PPW-2/WAGO-3, but is instead found at amino acid 7. This may provide for some level of proteolysis of PPW-2/WAGO-3 that renders it functional. Our results with N-terminally tagged PPW- 2/WAGO-3 are comparable to those of Schreier et al (Nat Cell Biol 2021), who inserted a similar tag at the N-terminus:

“we endogenously tagged WAGO-3 with GFP::3×FLAG at its N terminus. We note that this may affect the function of WAGO-3, as its N terminus is processed34. However, disruption of N-terminal WAGO-3 processing does not result in phenotypes in WAGO-1-wild-type animals34, as is the case in our experiments.”

We also overlapped our PPW-2/WAGO-3 IP/small RNA enrichment data with that of Schreier et al. and Gudipati et al. (Gudipati tagged PPW-2/WAGO-3 internally, after aa 46) and found good correlation with both data sets, suggesting that the N terminal tagging of PPW-2/WAGO-3 does not interfere with small RNA association in an otherwise wild-type background.

To increase transparency in our results, we have added the following sentence in the first results section to address the potential caveats of N-terminal tagging of PPW-2/WAGO-3 and WAGO-1:

We note that although PPW-2 (also known as WAGO-3) and WAGO-1 were recently shown to be N-terminally processed by the protease DPF-1, both GFP::3xFLAG::PPW-2 and 3xFLAG::WAGO-1 strains behaved as wild-type, consistent with previous reports (Gudipati et al., 2021; Schreier et al., 2022).